# Federated Class-Incremental Learning with Hierarchical Generative Prototypes

**Riccardo Salami** *                        *riccardo.salami@unimore.it*
**Pietro Buzzega** *                         *pietro.buzzega@unimore.it*
**Matteo Mosconi**                           *matteo.mosconi@unimore.it*
**Mattia Verasani**                          *mattia.verasani@unimore.it*
**Simone Calderara**                         *simone.calderara@unimore.it*
*AImageLab, University of Modena and Reggio Emilia*

**Reviewed on OpenReview:** *https://openreview.net/forum?id=k2TT42Ei8W*

## Abstract

Federated Learning (FL) aims at unburdening the training of deep models by distributing computation across multiple devices (clients) while safeguarding data privacy. On top of that, Federated Continual Learning (FCL) also accounts for data distribution evolving over time, mirroring the dynamic nature of real-world environments. While previous studies have identified Catastrophic Forgetting and Client Drift as major factors of performance degradation in FCL, we shed light on the importance of *Incremental Bias* and *Federated Bias*, which cause models to prioritize classes that are recently introduced or locally predominant, respectively. Our proposal constrains both biases to the last layer by efficiently fine-tuning a pre-trained backbone using learnable prompts, resulting in clients that produce less biased representations and more biased classifiers. Therefore, instead of solely relying on parameter aggregation, we leverage generative prototypes to effectively balance the predictions of the global model. Our proposed methodology significantly improves the current state of the art across six datasets, each including three different scenarios[1].

## 1 Introduction

The traditional paradigm in Deep Learning necessitates accessing large-scale datasets all at once, which hinders scalability and raises privacy concerns, especially when sensitive data is involved. Although distributing training could be a solution, there is still no effective mechanism for blending trained models into a single unified one. Federated Learning (FL) (McMahan et al., 2017) addresses this challenge through a centralized server coordinating distributed devices to create a unified model while minimizing communication cost.

Federated Class-Incremental Learning (FCIL) (Yoon et al., 2021; Dong et al., 2022; Zhang et al., 2023b) takes a step further and couples distributed training with Online Learning, tolerating distribution shifts in the data over time. This presents new challenges, as deep models learning online (without relying on old examples) experience severe performance degradation due to Catastrophic Forgetting (McCloskey & Cohen, 1989). In FCIL, the training process unfolds in tasks, each of which shifts the data distribution by introducing new categories. Each task is divided into communication rounds, wherein the local models train on their private data distribution. After local training, each client may transmit information to the orchestrator (server), which creates a global model and redistributes it to all clients. In the literature, some methodologies account for architectural heterogeneity (*i.e.*, heterogeneous FL (Diao et al., 2021; Kim et al., 2022b; Ilhan et al., 2023)), while others aim to enhance the performance of local models without necessarily converging to a global one (*i.e.*, personalized FL (Collins et al., 2021; Ma et al., 2022; OH et al., 2022)). Our objective is instead to collaboratively train a unified global model across a distributed system, as in Dong et al. (2022).

---

*Equal contribution.

[1]Our codebase to reproduce the results is available at https://github.com/aimagelab/fed-mammoth.

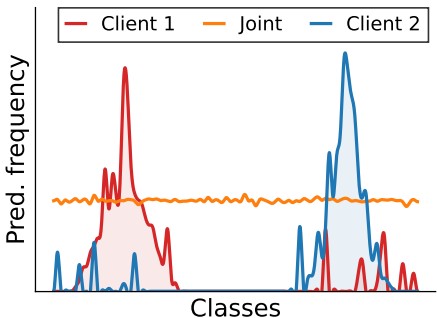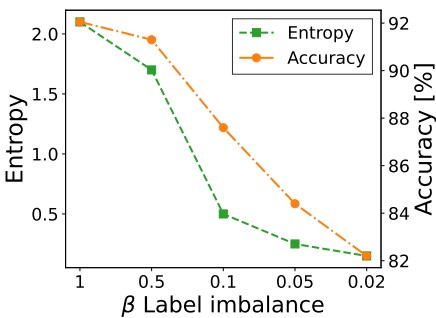

Figure 1: **Federated bias.** Histogram of the clients' responses on the global test set ($\beta = 0.05$) (left). Entropy of the response histograms, averaged on all clients, compared with FL performance (right).

When learning on a sequence of tasks, the model struggles the most at differentiating classes from distinct tasks, whereas it works well at separating those within the same one. Albeit one would intuitively link such behavior to Catastrophic Forgetting, it primarily occurs because tasks are learned separately, and some classes are never seen simultaneously (Kim et al., 2022a). This phenomenon is known in the Incremental Learning literature as *bias*, and emerges because new classification heads are optimized independently, without concurrent access to previous classes (Wu et al., 2019). This creates high gradient on the new classes, favoring them and causing imbalance in the classifier's output. We refer to it as Incremental Bias (IB) in this work.

The authors of Luo et al. (2021) first observed and formalized a similar tendency in Federated Learning: as clients train exclusively on their local datasets, they exhibit a *bias* towards their local label distribution. We refer to this effect as *Federated Bias* (FB) in our work. In contrast to the well-known Client drift (Gao et al., 2022; Karimireddy et al., 2020; Zhao et al., 2018), which causes misalignment between the clients' learned *parameters*, Federated Bias affects the clients' *responses*. Specifically, FB induces clients' outputs to diverge in different directions, mirroring the patterns of their local label distributions. Also, its strength increases with heterogeneity, suggesting a correlation with declining performance (see Section 2). To relieve such an effect, we constrain FB to the last layer by fine-tuning a frozen pre-trained backbone via prompt learning (Li & Liang, 2021). Ideally, prompting learns small updates, keeping clients' representations close to the pre-training (hence, close to each other); we verify this intuition empirically in Section 4.3. This confines the impact of FB to the last layer, providing the centralized server with less biased representations. On top of that, prompt-based methods *i)* provide SOTA results in Class-Incremental Learning (Wang et al., 2022b;a; Smith et al., 2023) and *ii)* adapt only a small portion of the clients' parameters, improving communication efficiency in distributed scenarios (Zhao et al., 2023; Liu et al., 2023).

The authors of Wu et al. (2019); Zhang et al. (2023a); Luo et al. (2021) address either IB or FB by fine-tuning the classification layer (where such biases are the most evident) on IID data samples. To meet the privacy requirements of FL, which prohibit transferring real data, we follow recent studies (Zhang et al., 2023a; Luo et al., 2021) and leverage latent generative replay. Specifically, at each communication round, we alleviate both biases – previously enforced to the last layer by the adopted prompt-based fine-tuning – by rebalancing the global classifier on a dataset of generated representations. In contrast to other approaches relying on prototypes (*i.e.*, the average feature vectors) to regularize clients' training procedures (Tan et al., 2022a; Guo et al., 2025), we propose to compute their covariance matrix and parameterize a Multivariate Gaussian distribution for each class-client combination. This forms a grid of *num_classes × num_clients generative prototypes*, which are sampled hierarchically (first by class, then by client) to generate new data points.

Summarizing, this work:

- sheds light on the relation between prompt learning and the aforementioned biases, identifying the latter as a major factor of performance degradation in FCIL;
- proposes a novel methodology that confines (with prompting) and mitigates (by rebalancing) such biases in the final classification layer;
- provides a comprehensive evaluation of the proposed approach, demonstrating state-of-the-art performance on standard benchmarks with low communication costs.

## 2    On Federated Bias

This section investigates how Federated Bias in clients' responses is associated with performance degradation in Federated Learning. All experiments are conducted on the CIFAR-100 (Krizhevsky et al., 2009) dataset, where the data is heterogeneously distributed across 10 clients under the commonly adopted distribution-based label imbalance setting (Li et al., 2022; Yurochkin et al., 2019).

To effectively show the presence of FB in the local models, we evaluate them at their most biased state: specifically, immediately after completing local training and before any synchronization with the central server. This snapshot removes aggregation effects and reveals the client-specific skew induced by non-IID data. In Figure 1 (left), we show the histogram of predicted labels on the global test set for two randomly selected clients trained with $\beta = 0.05$, and compare it with a model trained centrally on the full data distribution (referred to as Joint). Notably, while the Joint model aligns with the global class prior as expected, the clients' predicted label distributions are skewed toward prevalent classes in their datasets, confirming the presence of pronounced FB in local models.

To define a quantitative measure for FB, we consider the responses from all clients and compute the average entropy of the histograms of their predictions. Here, low entropy indicates a highly biased model presenting a peaked response distribution, whereas high entropy implies uniformity in the model's responses and is linked to lower bias. The experiment is repeated for five increasingly challenging label-imbalance settings ($\beta \in \{1.0, 0.5, 0.1, 0.05, 0.02\}$). Figure 1 (right) shows the average entropy at the end of the local training compared to the final performance. The two curves are notably similar, suggesting a correlation between Federated Bias and performance deterioration. Results on more datasets can be found in Section F.

## 3    Methodology

In this section, we formalize the Federated Class-Incremental Learning (FCIL) setting and introduce Hierarchical Generative Prototypes (HGP), coupling a single shared prompt in early attention blocks with server-side classifier rebalancing using hierarchical generative prototype.

### 3.1    Problem definition

Federated Class-Incremental Learning (Zhang et al., 2023b; Guo et al., 2025) tackles a classification problem across $C$ classes, which are introduced sequentially over $T$ incremental tasks. For each task, the data is distributed in a non-IID manner among $M$ clients. Let $D^t$ be the global partition for task $t$, which is split among the $M$ clients, with $D_m^t$ being the local partition of client $m$ at task $t$. The training procedure of each task is divided into communication rounds, each consisting of a certain number of epochs. At the end of the local optimization, the clients synchronize with the server by exchanging their learnable parameters. The server aggregates these parameters into a global model and redistributes it to all clients, thus concluding the communication round. Since data from previous tasks is unavailable, client $m$ can only train on its dataset $D_m^t$ during task $t$. Letting $f_{\theta_m}$ be the local model of client $m$, parameterized by $\theta_m$, the local objective for each client is to minimize the loss function $\mathcal{L}$ with respect to its local dataset $D_m^t$, namely:

$$\underset{\theta_m}{\text{minimize}} \quad \mathbb{E}_{(x,y) \sim D_m^t} \mathcal{L}\left(f_{\theta_m}(x), y\right). \tag{1}$$

The goal for the centralized server is to find the optimal set of parameters $\theta$ that minimizes the loss function on the entire dataset, without having access to any data point:

$$\underset{\theta}{\text{minimize}} \quad \frac{1}{TM} \sum_{t=1}^{T} \sum_{m=1}^{M} \mathbb{E}_{(x,y) \sim D_m^t} \mathcal{L}\left(f_{\theta}(x), y\right). \tag{2}$$

### 3.2    Hierarchical Generative Prototypes

We introduce Hierarchical Generative Prototypes (HGP), which comprises two key components: i) *prompting*, reducing communication costs and constraining Federated and Incremental biases within the classifier; ii) *classifier rebalancing*, addressing these biases on the server side.

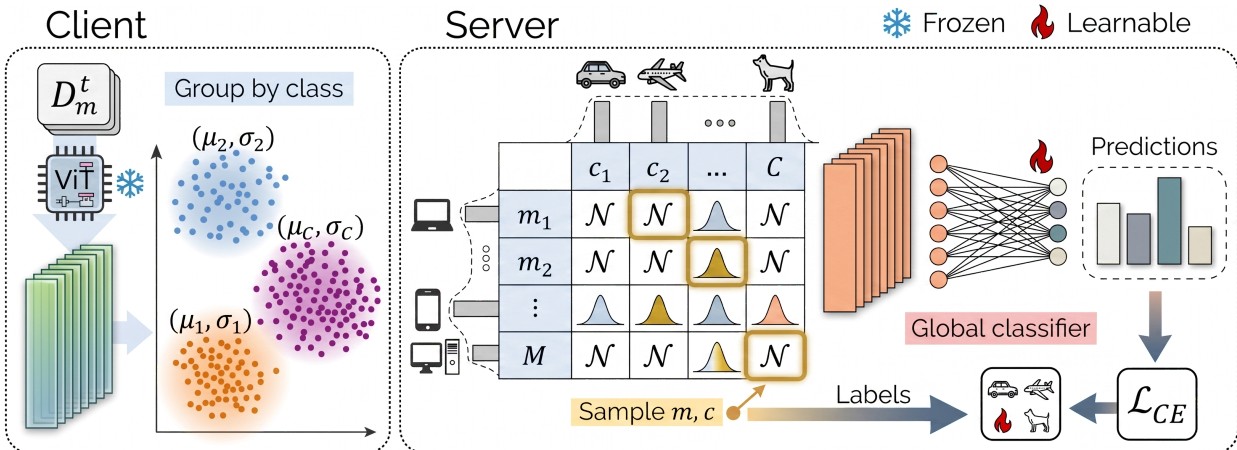

Figure 2: Classifier Rebalancing procedure through hierarchical sampling.

*Prompting.* Differently from other prompt-based CL approaches, which rely on a *pool* of task-generic (Wang et al., 2022b) or task-specific (Smith et al., 2023; Wang et al., 2022a) prompts, we propose learning a *single* prompt shared across all tasks. This eliminates the need for prompt selection – which requires an additional forward pass – significantly accelerating both training and inference. Specifically, we instantiate two learnable vectors, $P_k$ and $P_v$, for each of the first 5 transformer blocks, obtaining the prompt $\mathcal{P} = \{P_k^j, P_v^j \mid j \in 1, \dots, 5\}$. Following prefix-tuning (Li & Liang, 2021), $P_k^j$ and $P_v^j$ are respectively prepended to the keys and values of the $j^{th}$ Multi-head Self Attention layer and optimized jointly to minimize the loss function. Unlike prompt-pool methods that instantiate task-specific prompts to avoid interference, we use a single prompt with higher capacity, which allows accumulating task knowledge without routing. We verify in Section G that our design mitigates catastrophic forgetting.

After training on the local data distribution, each client sends its learnable parameters (consisting of prompt and classification head) to the server. Then, given $|D^t| = \sum_{m=1}^{M} |D_m^t|$, where $|\cdot|$ represents the cardinality of a set, the server aggregates the parameters as:

$$\theta^t = \frac{1}{|D^t|} \sum_{m=1}^{M} |D_m^t| \, \theta_m^t. \tag{3}$$

Here, $\theta_m^t = \{\mathcal{P}_m, W_m^t\}$ indicate the learnable parameters of client $m$ during task $t$ (comprising of prompt $\mathcal{P}_m$ and classification head $W_m^t$).

*Classifier Rebalancing.* To mitigate both biases in the classification head, we employ a rebalancing procedure on the server side. Specifically, we retrain the final classification layer using data entries sampled from *generative prototypes.*

The procedure starts with each client $m$ approximating the distribution of the features $h$ related to class $c$ using a multivariate Gaussian $\mathcal{N}_{m,c}(\mu_{m,c}, \Sigma_{m,c})$, where $\mu_{m,c}$ is the mean and $\Sigma_{m,c}$ the covariance matrix of the feature vectors produced by its local examples belonging to class $c$ (see Figure 2, Client). Since other studies denote $\mu_{m,c}$ as the prototype for client $m$ and class $c$, we refer to $\mathcal{N}_{m,c}$ as a generative prototype, given its capability to synthesize new samples. By repeating this process across all classes and clients, we produce $M \times C$ prototypes, which need to be combined into a single generative model that best approximates the global data distribution.

For a given class $c$, the server receives $M$ generative prototypes, one from each client, and seeks to identify the single distribution that best aligns with all of them. To this aim, we take the distribution that minimizes the Jensen-Shannon Divergence (JSD) (Lin, 1991). Considering multiple distributions $\mathcal{Q} = \{Q_1, \dots, Q_M\}$,

Table 1: **CIFAR-100, ImageNet-R and ImageNet-A**. Results in terms of FAA [↑]. Best are highlighted in bold, second-best underlined.

| | CIFAR-100 | | | ImageNet-R | | | ImageNet-A | | |
|---|---|---|---|---|---|---|---|---|---|
| **Joint** | **92.75** | | | **84.02** | | | **54.64** | | |
| **Partition** $\beta$ | 0.5 | 0.1 | 0.05 | 0.5 | 0.1 | 0.05 | 1.0 | 0.5 | 0.2 |
| EWC | 78.46 | 72.42 | 64.51 | 58.93 | 48.15 | 43.68 | 10.86 | 10.07 | 8.89 |
| LwF | 62.87 | 55.56 | 47.09 | 54.03 | 41.02 | 46.07 | 8.89 | 8.89 | 7.90 |
| FisherAVG | 76.10 | 74.43 | 65.31 | 58.68 | 50.82 | 47.33 | 11.59 | 11.06 | 10.14 |
| RegMean | 59.80 | 45.88 | 39.08 | 61.18 | 57.00 | 55.80 | 8.56 | 6.22 | 4.34 |
| CCVR | 79.95 | 75.14 | 65.30 | 70.00 | 62.60 | 60.38 | 39.50 | 36.27 | 35.94 |
| L2P | 83.88 | 61.54 | 55.00 | 42.08 | 23.85 | 16.98 | 20.14 | 17.31 | 16.85 |
| DualPrompt | 80.49 | 54.31 | 42.43 | 45.89 | 27.34 | 21.76 | 20.91 | 16.50 | 9.02 |
| CODA-P | 82.25 | 61.82 | 46.74 | 61.18 | 36.73 | 25.82 | 18.30 | 14.48 | 7.31 |
| FedProto | 75.79 | 70.02 | 60.55 | 58.52 | 47.30 | 52.93 | 9.87 | 9.22 | 10.01 |
| TARGET | 74.72 | 72.32 | 62.60 | 54.65 | 45.83 | 41.32 | 10.27 | 11.39 | 10.73 |
| PIP | 85.01 | 81.87 | 81.40 | 58.74 | 55.52 | 56.64 | 26.90 | 26.38 | 24.69 |
| PILoRA | 76.48 | 75.81 | 74.80 | 53.67 | 51.62 | 49.37 | 19.62 | 18.70 | 20.01 |
| LoRM | 86.95 | 81.75 | 82.76 | 72.48 | 63.83 | 66.45 | 37.26 | 36.34 | 33.11 |
| **HGP** (ours) | **90.39** | **90.20** | **90.16** | **72.64** | **71.93** | **71.58** | **41.61** | **41.01** | **40.62** |

with importance weights $\Pi = \{\pi_1, \ldots, \pi_M\}$, their JSD is defined as:

$$\mathrm{JSD}_\Pi(\mathcal{Q}) = \sum_{m=1}^M \pi_m D_{\mathrm{KL}}\left(Q_m || G\right), \quad G = \sum_{m=1}^M \pi_m Q_m, \tag{4}$$

where $D_{\mathrm{KL}}$ is the Kullback-Leibler divergence. Therefore, our objective is to find a global distribution $\tilde{Q}_c$ for the class $c$ that optimizes the following objective:

$$\underset{\tilde{Q}_c}{\mathrm{minimize}} \quad \sum_{m=1}^M \pi_{m,c} D_{\mathrm{KL}}\left(\tilde{Q}_c \, || \, G\right), \tag{5}$$

Notably, the optimal solution to this problem is to set $\tilde{Q}_c$ equal to the distribution $G$. Since, in our case, all distributions $\{Q_1, \ldots, Q_M\}$ are Gaussians, the closest distribution $G$ is precisely defined as a Gaussian Mixture Model (GMM). Following a common practice in Federated Learning (Tan et al., 2022a; McMahan et al., 2017), we assign the importance weights $\{\pi_{1,c}, \ldots, \pi_{M,c}\}$ to the generative prototypes based on the number of samples from class $c$ observed by client $m$.

By solving Equation 5 for all classes, we result in $C$ GMMs, $\{\tilde{Q}_1, \ldots, \tilde{Q}_C\}$, one for each class. To obtain a single generative model, we further combine them by following a similar reasoning to the one of Equation 5. This time, we consider those $C$ class-specific GMMs as the initial distributions in Equation 4, which leads us to define the global generative model $\tilde{Q}$ over all classes:

$$\tilde{Q} = \sum_{c=1}^C \omega_c \tilde{Q}_c, \quad \tilde{Q}_c \triangleq \sum_{m=1}^M \pi_{m,c} \mathcal{N}_{m,c}\left(\mu_{m,c}, \Sigma_{m,c}\right), \tag{6}$$

where the class-specific weights $\{\omega_1, \ldots, \omega_C\}$ are set to the normalized number of samples for each class. According to Equation 5, $\tilde{Q}$ is the distribution that most closely aligns with all class-specific GMMs: consequently, it also provides the closest match to all the initial Gaussians across every class-client combination.

When it comes to sampling from $\tilde{Q}$, we recall that the standard GMM sampling process consists of two steps: first, selecting a Gaussian, and then sampling from it. Expanding on this framework, we introduce an

---

**Algorithm 1** Hierarchical Generative Prototypes **HGP**

---

1: **Input:** generic task $t$; $M$ clients; local model $f_{\theta_m^t}(\cdot)$ parameterized by $\theta_m^t = \{\mathcal{P}_m, W_m^t\}$; $N$ prompts; $C$ classes; $E$ local epochs; $E_r$ rebalancing epochs; local learning rate $\eta$; rebalancing learning rate $\eta_r$.
2: **for each** communication round **do**
3:    **Server side:**
4:    Server distributes $\theta^t$
5:    **Client side:**
6:    **for each** client $m \in \{1, \ldots, M\}$ **in parallel do**
7:       $\theta_m^t = \theta^t$
8:       **for each** local epoch $e \in \{1, \ldots, E\}$ **do**
9:          **for each** batch $(x, y) \sim D_m^t$ **do**
10:             $\theta_m^t \leftarrow \theta_m^t - \eta \nabla \mathcal{L}_{\mathrm{CE}}(f_{\theta_m^t}(x), y)$
11:       Compute feature vectors $h$ on $D_m^t$
12:       **for each** class $c \in C^t$ **do**
13:          Compute $\mu_{m,c}$ and $\Sigma_{m,c}$ to parameterize $\mathcal{N}_{m,c}$
14:       Send parameters $\theta_m^t$ and Gaussians $\mathcal{N}_{m,c}$ to the server
15:    **Server side:**
16:    Compute $\theta^t = \dfrac{1}{|D^t|} \sum_{m=1}^M |D_m^t| \, \theta_m^t$
17:    **for each** class $c \in C^t$ **do**
18:       Compute $\tilde{Q}_c = \sum_{m=1}^M \pi_{m,c} \mathcal{N}_{m,c}(\mu_{m,c}, \Sigma_{m,c})$
19:    Compute $\tilde{Q} = \sum_{c=1}^C \omega_c \tilde{Q}_c$
20:    Collect dataset $\hat{D}$ sampling feature vectors from $\tilde{Q}$
21:    **for each** rebalancing epoch $e \in \{1, \ldots, E_r\}$ **do**
22:       Optimize $W$ using Equation 7

---

additional step, which is used to select the class-specific GMM $\tilde{Q}_c$ from the global distribution $\tilde{Q}$. Specifically, as illustrated in Figure 2 (Server), our hierarchical sampling strategy draws: *i)* the class-specific GMM index ($c$) from a Multinomial distribution on the class weights $\{\omega_1, \ldots, \omega_C\}$; *ii)* the client-specific Gaussian index ($m$) from a Multinomial distribution on the class-client weights $\{\pi_{1,c}, \ldots, \pi_{M,c}\}$; *iii)* the synthetic feature $\hat{h}$ from the generative prototype $\mathcal{N}_{m,c}$.

After each communication round, the server (i) generates a synthetic dataset $\hat{D}$ by sampling from $\tilde{Q}$, (ii) aggregates clients' learnable parameters $\theta_m^t$ via Equation 3, and (iii) rebalances the global classifier G (parameterized by $W$) by minimizing the Cross-Entropy (CE) loss on $\hat{D}$:

$$\underset{W}{\mathrm{minimize}} \; \mathbb{E}_{(\hat{h}, c) \sim \hat{D}} \mathcal{L}_{\mathrm{CE}} \left( G_W(\hat{h}), c \right). \tag{7}$$

Finally, the server redistributes the updated classifier along with the aggregated prompt to the clients, enabling them to start the next training round. We report the pseudo-code for a generic task $t$ in Algorithm 1, and a detailed analysis of the computational complexity of our approach in Section 4.5.

## 4 Experiments

In this section, we assess the effectiveness of the proposed method, comparing it with the current State of The Art in Federated Class-Incremental Learning.

### 4.1 Setting

*Datasets.* Following (Salami et al., 2025), we evaluated the proposed method on *six* diverse datasets, spanning both in-distribution and out-of-distribution domains: *CIFAR-100* (Krizhevsky et al., 2009), *ImageNet-R* (Hendrycks et al., 2021a), *ImageNet-A* (Hendrycks et al., 2021b), *EuroSAT* (Helber et al., 2018), *Cars-196* (Krause et al., 2013), and *CUB-200* (Wah et al., 2011). All datasets are partitioned into 10 incremental

Table 2: **EuroSAT, Cars-196 and CUB-200**. Results in terms of FAA [↑]. Best are highlighted in bold, second-best underlined.

| | EuroSAT | | | Cars-196 | | | CUB-200 | | |
|---|---|---|---|---|---|---|---|---|---|
| Joint | 98.42 | | | 85.62 | | | 86.04 | | |
| Partition $\beta$ | 1.0 | 0.5 | 0.2 | 1.0 | 0.5 | 0.2 | 1.0 | 0.5 | 0.2 |
| EWC | 64.12 | 59.30 | 56.52 | 19.55 | 18.02 | 18.29 | 31.46 | 29.60 | 27.89 |
| LwF | 31.91 | 21.26 | 31.42 | 20.84 | 22.72 | 31.76 | 25.25 | 21.11 | 18.54 |
| FisherAVG | 58.84 | 59.94 | 55.86 | 26.03 | 24.60 | 21.58 | 30.45 | 28.39 | 25.06 |
| RegMean | 48.74 | 51.73 | 45.27 | 21.83 | 20.36 | 15.92 | 35.57 | 32.84 | 32.83 |
| CCVR | 64.44 | 57.93 | 62.69 | 38.99 | 37.81 | 35.31 | 62.67 | 59.48 | 56.33 |
| L2P | 40.63 | 51.78 | 45.46 | 35.49 | 31.00 | 20.01 | 56.23 | 47.31 | 38.16 |
| DualPrompt | 62.97 | 52.78 | 55.34 | 34.07 | 26.47 | 21.30 | 60.93 | 55.59 | 44.61 |
| CODA-P | 73.38 | 69.42 | 66.69 | 28.04 | 20.83 | 14.53 | 42.53 | 37.71 | 29.19 |
| FedProto | 58.79 | 62.85 | 64.17 | 26.08 | 24.55 | 22.75 | 30.22 | 28.27 | 26.01 |
| TARGET | 52.74 | 52.74 | 45.11 | 28.65 | 27.20 | 26.13 | 39.30 | 38.40 | 34.79 |
| PIP | 58.12 | 53.67 | 54.16 | 36.02 | 34.33 | 29.99 | 69.60 | 65.46 | 60.79 |
| PILoRA | 48.35 | 32.89 | 31.22 | 37.57 | 37.92 | 36.95 | 61.11 | 60.68 | 60.39 |
| LoRM | 84.23 | 77.26 | 81.36 | **54.41** | **51.87** | 48.81 | 64.60 | 63.67 | 60.06 |
| **HGP** (ours) | **87.97** | **88.35** | **85.90** | 52.13 | 51.48 | **49.95** | **80.23** | **79.71** | **78.58** |

tasks, except for *EuroSAT*, which is split into 5 tasks. We distribute the data across 10 clients using the widely adopted *distribution-based* label imbalance setting (Li et al., 2022; Yurochkin et al., 2019). This approach partitions the data according to a Dirichlet distribution controlled by a $\beta$ parameter. Additional discussion involving up to 100 clients are provided in Section C.1. For each dataset, we experiment with three different values of $\beta$, selected based on the number of examples. Specifically, we use $\beta \in \{0.5, 0.1, 0.05\}$ for ImageNet-R and CIFAR-100, and $\beta \in \{1.0, 0.5, 0.2\}$ for EuroSAT, ImageNet-A, Cars-196 and CUB-200. This choice is imposed by the infeasibility of finding a valid split when the data samples or the number of classes are limited. We evaluate the centralized model at the end of the training on the global test set.

*Implementation Details.* We use a pre-trained ViT-B/16 as the backbone for all compared methods, initializing the models with supervised pre-trained weights on ImageNet-21K (Ridnik et al., 2021). For a comprehensive overview and a sensitivity analysis of the hyperparameters please refer to the Appendix. The results are averaged over three runs on different seeds. We report the standard deviations in Section E.

## 4.2 Results

*Metrics.* We assess the performance of all methods using the widely adopted Final Average Accuracy (FAA). For the formal definition of such measure, we refer the reader to Section C.

*Evaluated approaches.* Our proposal is evaluated alongside 13 competitors. Among these, *five* were originally designed for Class-Incremental Learning (CIL), *two* for Federated Learning (FL), *two* for Model Merging, and the remainder *four* for Federated Class-Incremental Learning (FCIL). Following common practice, we adapt CIL approaches for FCIL by aggregating local models with FedAvg (McMahan et al., 2017). From CIL, we include two regularization-based techniques (EWC (Kirkpatrick et al., 2017), LwF (Li & Hoiem, 2017)) and three prompting-based approaches (L2P (Wang et al., 2022b), DualPrompt (Wang et al., 2022a), and CODA-Prompt (Smith et al., 2023)). From FL, we select a rebalancing method (CCVR (Luo et al., 2021)), and one using a prototype-based classifier (FedProto (Tan et al., 2022a)); we include two model merging techniques (FisherAVG (Matena & Raffel, 2022) and RegMean (Jin et al., 2023)), and four FCIL methods. These leverage Parameter-Efficient Fine-Tuning (PILoRA (Guo et al., 2025)), generative replay (TARGET (Zhang et al., 2023b)), regularization prototypes (PIP (Ma'sum et al., 2024)), and closed-form

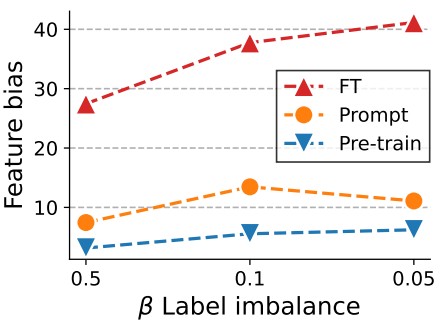 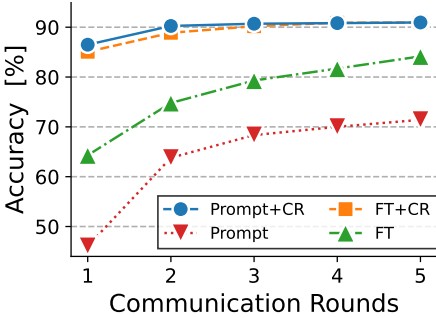

Figure 3: **Prompting vs. fine-tuning.** Average pairwise distance of the local prototypes on all clients (left). FL performance before and after Classifier Rebalancing (CR), for $\beta = 0.05$ (right).

merging (LoRM (Salami et al., 2025)). Finally, we include the upper bound, where the same ViT-B/16 backbone is trained *jointly* on the global data distribution (*i.e.*, no dataset splitting), referred to as "Joint".

*Comparison.* In Table 1, we present the results for CIFAR-100, ImageNet-R, and ImageNet-A. Consistently on the first two datasets, prompting methodologies struggle as the label heterogeneity in the data distribution increases. This aligns with our findings, suggesting that federated bias is more pronounced when learning prompts (see Section 4.3). In contrast, FisherAVG and LwF are steadier, but still exhibit poor performance overall. EWC and FisherAVG surprisingly perform similar to FL methodologies (CCVR, FedProto), only to be surpassed by FCIL-targeted approaches (PILoRA, PIP, TARGET, LoRM, HGP).

ImageNet-A, designed to evaluate model robustness, poses a greater challenge by incorporating samples from ImageNet that are frequently misclassified. This is also evidenced by the lower performance of all methods. Here, classical CL techniques struggle to learn the task effectively and are surpassed by prompting methodologies. Among FL and FCIL approaches, only PIP and PILoRA achieve decent performance, with CCVR and LoRM outperforming them. Finally, thanks to the combination of classifier rebalancing and prompting, HGP consistently delivers the best performance in all settings, while LoRM ranks as the second-best performer on average.

Table 2 compares the same methodologies on EuroSAT, Cars-196, and CUB-200. Among these datasets, Cars-196 is the most impacted by the FCIL setting, as it exhibits the largest performance gap between joint training and the best-performing method. In this scenario, EWC and LwF fail to deliver satisfactory results, whereas FL and FCIL methodologies achieve competitive performance. While rebalancing played a critical role in the previously analyzed datasets, CCVR performs slightly worse than PILoRA in this case, highlighting the significant impact of low-rank adaptation. Here, HGP and LoRM compete for the top spot by a wide margin over the others, showing comparable performance.

For EuroSAT and CUB-200, the accuracy gap between Joint training and FCIL techniques narrows, suggesting a less challenging task. FedProto, CODA-Prompt, and DualPrompt demonstrate superior results on EuroSAT, whereas PIP, PILoRA and CCVR underperform on this dataset but achieve the best results on CUB-200. This trend reversal is likely due to the larger number of classes in CUB-200, which challenges traditional CL methods and highlights the significance of approaches tailored for Federated Learning. LoRM emerges as the clear second-best performer on EuroSAT and struggles on CUB-200, where its performance aligns with that of PIP and PILoRA. HGP consistently delivers the highest performance across both datasets, with a large margin on CUB-200. See per-task accuracy curves in Section G.

## 4.3 Ablation study

*Prompting vs. fine-tuning.* In the following, we examine how prompting influences Federated Bias (FB), comparing it directly to conventional full-network fine-tuning[2]. We argue that prompting is a more advantageous approach for adapting pre-trained models in Federated Learning, as it constrains the bias to

---

[2]All experiments are performed on CIFAR-100 distributed across 10 clients.

the last layer instead of distributing it across the whole network. To experimentally prove this claim, we position ourselves at the end of the local training prior to the first synchronization with the server: namely, when each client is trained exclusively on its local distribution. In this scenario, we assess the Federated Bias within the feature space – which we refer to as *feature bias* for short – by computing the local prototypes for the 10 clients and measuring their average *pairwise Euclidean distance*. Figure 3 (left) shows that prompting produces smaller feature bias compared to traditional fine-tuning, suggesting that the obtained features are less biased towards the local clients' distributions. To validate this assertion, we also compute the same metric on the pre-trained model (ViT-B/16, on ImageNet-21k), which yields the smallest bias by design. This further supports our reasoning, as this last network experiences no fine-tuning whatsoever.

On the right-hand side of Figure 3, we show the performance of prompting *vs.* traditional fine-tuning. In line with recent works (Zhang et al., 2023a; Panos et al., 2023; McDonnell et al., 2024), prompting (Prompt) shows inferior performance compared to fine-tuning (FT), due to its limited plasticity. However, when leveraging Classifier Rebalancing (CR), we observe a reversal in this trend. Simply addressing Federated Bias in the last layer is sufficient for prompting to outperform classical fine-tuning: this suggests that prompting techniques train more biased classifiers while producing less biased features, effectively restricting FB to the final layer.

*Impact of Different Components.* We assess the specific contributions of various components of HGP in terms of FAA for three datasets under the most challenging distribution-based label imbalance setting ($\beta = 0.05$). Results of these ablative experiments are summarized in Table 3, where the full fine-tuning of ViT-B/16 serves as the lower bound. Incorporating our prompting technique yields a remarkable improvement on CIFAR-100 and ImageNet-R, with a minor gain on CUB-200. However, the primary advan-

Table 3: **HGP components**. Evaluation of the impact of each component of HGP. Presented in terms of FAA [↑].

| Prompt | $CR_{old}$ | $CR_{cur}$ | C-100 | IN-R | CUB-200 |
|--------|------------|------------|-------|------|---------|
| ✗ | ✗ | ✗ | 30.58 | 26.42 | 25.70 |
| ✗ | ✓ | ✗ | 62.78 | 59.42 | 61.09 |
| ✗ | ✗ | ✓ | 65.30 | 60.38 | 56.33 |
| ✗ | ✓ | ✓ | 82.52 | 64.91 | 70.59 |
| ✓ | ✗ | ✗ | 52.29 | 30.28 | 26.74 |
| ✓ | ✓ | ✗ | 81.91 | 66.70 | 69.51 |
| ✓ | ✗ | ✓ | 85.43 | 68.88 | 47.80 |
| ✓ | ✓ | ✓ | 90.16 | 71.58 | 78.58 |

tage of prompting lies not only in enhancing accuracy but also in confining both types of bias to the final classification layer, consequently increasing the impact of Classifier Rebalancing.

We define $CR_{cur}$ and $CR_{old}$ as classifier rebalancing procedures applied to the current task classes and the old task classes, respectively. Specifically, $CR_{cur}$ rebalances only the classification head for the current task by utilizing generated features from prototypes corresponding to the current classes. Conversely, $CR_{old}$ focuses on rebalancing the classification heads for previous tasks by sampling from the prototypes of the old classes. On average, both approaches perform comparably; however, $CR_{old}$ demonstrates superior performance on CUB-200, highlighting the importance of Incremental Bias in this dataset. By integrating all components, HGP take advantage of the contribution of each, achieving SOTA performance as shown in Tables 1 and 2.

## 4.4 Communication Cost

Efficient communication is paramount in Federated Learning because of the extensive coordination required between server and clients. In this section, we analyze the relationship between Final Average Accuracy (FAA) and communication cost for each method. The FAA results are provided for CUB-200, ImageNet-R, and CIFAR-100 (Figure 4), with $\beta = 0.05$. Communication costs (*i.e.*, the average data exchanged between a single client and the server) are reported in Megabytes (MB) per communication round.

Notably, methods that do not utilize PEFT techniques incur significantly higher communication costs, as they optimize all parameters. Consequently, with larger models, communication costs escalate proportionally without corresponding performance gains. TARGET is the most expensive method overall because the server sends the entire model and the generated dataset to clients once per task. In contrast, the adoption of PEFT techniques enhances both efficiency and performance. L2P emerges as the most efficient method, using fewer prompts than its competitors, followed by DualPrompt and CODA-P. PILoRA is on par to the previous

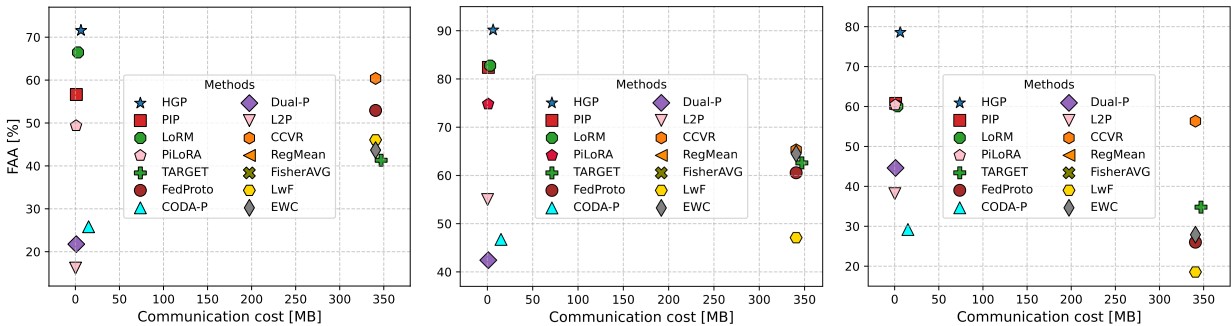

Figure 4: FAA [%] in relation with the communication cost [MB] for all tested approaches on ImageNet-R (left) and CIFAR-100 (center) and CUB-200 (right).

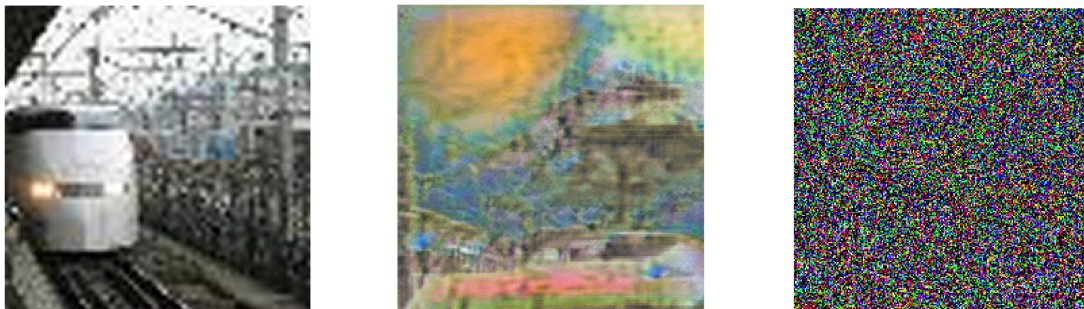

Figure 5: Real training image (left) and its reconstructions leveraging either the input image (center) or the generative prototype of the related class (right).

prompt-based solutions in terms of efficiency. HGP demonstrates efficiency comparable with the PEFT techniques, ensuring a strong performance gain. Overall, the efficiency gains and substantial performance improvements offered by PEFT techniques highlight their advantage in Federated Learning scenarios.

## 4.5 Computational Complexity

Our experimental setup aligns with prior work, assuming clients can fine-tune transformer-based networks. However, our method reduces training cost significantly by using prompt tuning, updating only ∼1.5% of the total model parameters.

To ensure efficiency and numerical stability, we employ diagonal covariance matrices (represented as vectors) as prototypes. Empirically, we validate representativeness of diagonal covariance in Section J, and show full covariance matrices provide no consistent gains and incur higher computation and memory costs in Section I.

Constructing and updating the hierarchical mixture model incurs negligible cost, as it only requires maintaining running statistics. Sampling from the mixture involves two categorical draws (for client and class selection) and one Gaussian draw (for synthetic features). Thanks to the Alias method (Walker, 1974), categorical sampling is performed in $\mathcal{O}(1)$, and Gaussian sampling in $\mathcal{O}(features\_dim)$.

In total, the sampling cost is approximately 770 FLOPs for ViT-B/16. For comparison, a single forward pass on ViT-B/16 requires ∼17 GFLOPs, making our sampling overhead negligible. Moreover, inference time is further reduced by our proposed prompting technique, which employs a single shared prompt rather than a prompt pool thus decreasing inference cost by up to 50%.

### 4.6 Prototypes of Features are Privacy-preserving

In Federated Learning, maintaining the privacy of local data distributions is of utmost importance. It is crucial that no private samples are directly transmitted from the client to the server. In the HGP framework, each client provides the server with a generative prototype for each observed class.

To verify the safety of generative prototypes, we firstly examine a hypothetical scenario where an attacker gains access to the trained model along with a real image utilized during training. In such a case, the attacker could potentially employ the feature inversion technique outlined in (Zhao et al., 2020). This methodology provides for training a autoencoder to precisely match the input image. Specifically, the autoencoder takes fixed noise as input and reconstructs the image. Then, both the real and reconstructed image are passed through the frozen backbone. The objective is to minimize the mean squared error (MSE) loss between their respective features. Figure 5 (left) shows the real training image from the Tiny-ImageNet dataset, while Figure 5 (center) displays the reconstruction obtained by the trained autoencoder. Through this process, no significant semantics can be recovered. At most, the class might be guessed, but the image does not provide any detail of the original sample.

In another setting, we suppose that the attacker gains access to the centralized server, which has visibility into clients' generative prototypes. To recover an input image, initial random noise in the input space is optimized by minimizing MSE between its features and those of a fixed sample from the generative prototype. Figure 5 (right) shows that this results in a shapeless, noisy reconstruction, making it impracticable for an attacker to recreate real images from these feature distributions. We refer the reader to Section H for quantitative results on membership-inference attacks.

## 5 Relation with prior works

*Federated Learning.* The traditional aggregation rule in Federated Learning is *weighted parameter averaging* (FedAvg) which combines client models at the end of each round proportionally to their local sample counts (McMahan et al., 2017). While simple and communication-efficient, FedAvg can suffer under strong data heterogeneity, where client updates drift away from a common optimum, slowing or destabilizing convergence. To mitigate the *client drift* phenomenon, subsequent methods regularize local training, constraining client updates to remain close to the centralized server. Among these, FedProx (Li et al., 2020) explicitly inject such regularization into the local objective, while SCAFFOLD (Karimireddy et al., 2020) introduces control variates to cancel client-specific gradient bias and improve alignment with the global objective. FedDC (Gao et al., 2022) estimates the local parameter shift and employs it as a correction term before aggregation, and GradMA (Luo et al., 2023) reroutes each update along a direction that jointly reduces the local loss while maintaining proximity to the server parameters.

On a different trajectory, CCVR (Luo et al., 2021) models each class with a Gaussian in feature space and synthesizes an IID feature set to calibrate the server-side classifier head. FedProto (Tan et al., 2022a) also computes per-client class prototypes but aggregates them into global vectors that serve as target representations in subsequent rounds, as do contrastive prototype-alignment methods (Mu et al., 2023; Tan et al., 2022b) that use them as discriminative anchors in the local loss. In contrast, our proposed methodology constructs a *hierarchical* Gaussian Mixture Model and samples from it to rebalance the classifier, while a single shared prompt confines both Federated and Incremental Bias to the final layer.

*Class-Incremental Learning.* Class-Incremental Learning (CIL) stands out as one of the most challenging settings within the Continual Learning domain (van de Ven et al., 2022). In this scenario, the training process is divided into sequential tasks, each introducing a distinct set of classes as the training progresses. To mitigate catastrophic forgetting in this scenario, early techniques introduce regularization that anchors the model to prior tasks, either by distilling predictions from earlier models (Li & Hoiem, 2017), or by penalizing parameter drift from previously learned checkpoints (Zenke et al., 2017; Kirkpatrick et al., 2017). Alternatively, rehearsal-based approaches involve storing samples in a limited memory buffer and replaying them to optimize either the original objective (Robins, 1995; Chaudhry et al., 2018) or a surrogate one based on knowledge distillation (Rebuffi et al., 2017; Buzzega et al., 2020). Lately, the emergence of pre-trained self-attentive architectures (Dosovitskiy et al., 2020) in the Computer Vision domain has paved the way for significant

advancements, particularly with the advent of Parameter-Efficient Fine-Tuning (PEFT) techniques. Recent approaches (Wang et al., 2022b;a; Smith et al., 2023) have harnessed prompting to achieve state-of-the-art performance in CIL, eliminating the need for a buffer to replay old samples. Instead, they employ a prompt pool comprising incrementally learned prompts, utilized to condition the network during the forward pass. In our approach, we adopt the prompting paradigm and introduce an efficient methodology that minimizes the computational overhead by eliminating the need for a double forward pass and enhancing performance.

*Federated Class-Incremental Learning.* The concept of Federated Class-Incremental Learning (FCIL) was initially introduced in (Yoon et al., 2021). In their work, the authors propose FedWeIT, which partitions client-side parameters into task-generic and task-specific components. To mitigate interference between clients, they implement sparse learnable masks to selectively extract relevant knowledge for each client. GLFC (Dong et al., 2022) takes a different approach by combining local buffers with class-aware gradient compensation loss. This strategy helps counteract catastrophic forgetting through rehearsal, while also adjusting the magnitude of gradient updates based on whether input samples belong to new or old classes. Building upon this framework, an enhanced version is introduced by the same authors in the LGA paper (Dong et al., 2023). TARGET (Zhang et al., 2023b) tackles forgetting by training a centralized generator network to produce synthetic data, maintaining a similar behavior to the generator used in previous tasks. The generative network populates a buffer after each task, allowing clients to utilize it for rehearsal. Recent advancements in FCIL involve fine-tuning pre-trained models using PEFT techniques. Fed-CPrompt (Bagwe et al., 2023) introduces a regularization term that encourages local prompts to diverge from global ones, enabling them to learn task-specific features. Instead, PILoRA (Guo et al., 2025) integrates LoRA (Hu et al., 2021) with prototypes, which are aggregated via a re-weighting mechanism on the server side, whereas LoRM (Salami et al., 2025) proposes a closed-form solution for merging parameter-efficient clients at the server side.

More recently, class-IL prompt learning strategies have been tailored to handle decentralized data streams. (Piao et al., 2024) introduce a prompt-based knowledge transfer mechanism across tasks and clients, while (Yu et al., 2024) propose learning prompts at multiple granularities to increase personalized performance. Other works like FedProK (Gao et al., 2024) ensure trustworthy incremental learning through prototypical feature transfer. Instead, FPPL (He et al., 2024) and PIP (Ma'sum et al., 2024) combine prompt learning with prototype aggregation, using local prompts alongside global prototypes. While these strategies mitigate interference, they rely on maintaining complex and multi-tiered prompt pools. Our approach avoids the double forward pass and management overhead of multi-granularity prompts by using a single shared prompt.

# 6 Conclusions

In this work, we propose Hierarchical Generative Prototypes (HGP), an approach aimed at mitigating Incremental and Federated Biases within Federated Class-Incremental Learning. HGP utilizes pre-trained architectures conditioned by prompting to achieve state-of-the-art performance while maintaining parameter efficiency. To tackle the problem of Federated Bias and Incremental Bias, we examine the implications of fine-tuning the entire model compared to using prompting techniques, demonstrating that the latter confines bias to the classification layer. Building on these insights, we propose Classifier Rebalancing – *i.e.*, sampling features from a hierarchical Gaussian Mixture Model to train the classifier across all observed classes – as an effective solution. Through the integration of prompt learning and Classifier Rebalancing, we achieve SOTA performance while learning only a minimal number of parameters.

## Broader Impact Statement

HGP targets privacy-preserving federated learning: clients transmit only class-conditional Gaussian statistics to avoid sharing raw data. Section 4.6 and Section H demonstrate that such statistics are resilient to feature-inversion and membership-inference attacks; nonetheless, transmitting class-conditional features distributions may still leak aggregate, distribution-level information about local data (*e.g.*, site-specific demographics in sensitive domains such as medical imaging). Deployments in these settings should consider complementary safeguards (*e.g.*, differential privacy on the shared statistics). We foresee no further negative societal impact beyond those common to federated and continual learning.

**Acknowledgments**

This article is the result of a collaboration with the partners of the STORE project (`https://edf-store.com`), funded by the European Defence Fund (EDF) under grant agreement EDF-2022-101121405-STORE.

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

## A    Limitations and Future Works.

This work builds upon Prompt Learning, which allows adapting a pre-trained architecture on a downstream task. Although leveraging a pre-trained model is a common strategy in Federated Learning literature (Guo et al., 2025; Liu et al., 2023; Bagwe et al., 2023), our findings may not extend to scenarios where models are trained from scratch. Moreover, although prompt tuning effectively localizes bias to the classifier layer, the behavior of other PEFT techniques – such as LoRA or Adapters – under continual decentralized scenarios remains unexplored. Investigating these methods may uncover important trade-offs between efficiency and bias control, which we leave for future work. Finally, assessing HGP's robustness to adversarial settings with malicious clients remains an important direction, as it is critical for ensuring reliable deployment in real-world applications. Finally, while HGP mitigates direct feature inversion, the transmission of generative prototypes may still leak distribution-level information of the original data.

## B    Implementation Details

We utilize a pre-trained ViT-B/16 as the backbone for HGP and all the compared methods. Specifically, we initialize the models with supervised pre-trained weights on ImageNet-21K (Ridnik et al., 2021) for all datasets.

In our method, we adopt a single prompt for all tasks, negating both the need for a double forward pass while also avoiding the need to store old task-specific parameters . Across all experiments, prompt components have a shape of $(200, d)$, where $d = 768$ represents the embedding dimension for the chosen architecture. We train said prompt using prefix tuning, conditioning the first 5 layers of the backbone. For each task, we perform five communication rounds, during which clients observe their local datasets for five epochs with a batch size of 16. Training is conducted using the Adam optimizer (Kingma & Ba, 2015) with a learning rate of 0.003; $\beta_1$ and $\beta_2$ are set to 0.9 and 0.999, respectively.

The centralized server rebalances the global classifier for five epochs. We use the SGD optimizer with a learning rate of 0.01 and a momentum of 0.9, with a batch size of 256, and also apply a cosine annealing learning rate scheduler. We generate an average of 256 feature vectors for each class encountered up to this point, multiplying the covariance associated with each prototype by 3 to enhance dataset diversity. All images are resized to $224 \times 224$ using bicubic interpolation and scaled to ensure their values fall within the range $[0, 1]$. Additionally, we employ random cropping and horizontal flipping as data augmentation techniques. All experiments are conducted on a single Nvidia RTX5000 GPU.

The specific training details for all implemented methods are documented in Section D.

## C    Metrics

We assess the performance of all methods using the most commonly used metric in FCIL literature: Final Average Accuracy (FAA).

FAA represents the mean accuracy on all observed tasks at the conclusion of the incremental training process. Mathematically, if $A_i^j$ denotes the accuracy on the $i^{th}$ task at the $j^{th}$ incremental step, where $i \leq j$, FAA can be expressed as:

$$\text{FAA} = \frac{1}{T} \sum_{i=1}^{T} A_i^T.$$

### C.1    Scalability to Large Client Populations

To assess the scalability of our approach, we conducted additional experiments on CIFAR-100 using 100 clients with varying participation rates under extreme distribution imbalance ($\beta = 0.05$). Specifically, we tested participation rates of 10%, 20%, 50%, and 100%, obtaining average accuracies of 80.82%, 87.33%, 89.51%, and 90.08%, respectively. These results demonstrate that our method remains robust even under the most challenging federated settings, with only 10 clients participating per round. Remarkably, in this

low-participation regime, our method still outperforms all competing methods evaluated with just 10 clients and full participation.

# D Hyperparameter Tables

The following acronyms and symbols are used throughout the paper: $lr$ denotes the learning rate; $lr_{pr}$ refers to the learning rate applied to prototypes; $\lambda_{KL}$ is the weighting factor for the Knowledge Distillation (KL) loss; $r$ represents the rank used in low-rank matrix approximations; $g_{ep}$ indicates the number of epochs used for training and sample generation in the generator network; and $\gamma$ is the decay coefficient applied to the off-diagonal elements of the Gram matrices for the backbone (first component) and classifier (second component). To ensure a fair comparison, we dynamically adjust each hyperparameter search range for each methodology so that the optimal configuration was always bracketed by other tested values.

## D.1 CIFAR-100

| $\beta$ | 0.5 | 0.1 | 0.05 |
|---|---|---|---|
| EwC | $lr$: 1e-5 | $lr$: 1e-5 | $lr$: 1e-5 |
| LwF | $lr$: 1e-5 | $lr$: 1e-5 | $lr$: 1e-5 |
| FisherAVG | $lr$: 1e-5 | $lr$: 1e-5 | $lr$: 1e-5 |
| RegMean | $lr$: 1e-5; $\gamma$: (0.5, 0.5) | $lr$: 1e-5; $\gamma$: (0.5, 0.5) | $lr$: 1e-5; $\gamma$: (0.5, 0.5) |
| CCVR | $lr$: 1e-5 | $lr$: 1e-5 | $lr$: 1e-5 |
| L2P | $lr$: 3e-2 | $lr$: 3e-2 | $lr$: 3e-2 |
| DualPrompt | $lr$: 5e-2 | $lr$: 5e-2 | $lr$: 5e-2 |
| CODA-P | $lr$: 1e-3 | $lr$: 1e-3 | $lr$: 1e-3 |
| FedProto | $lr$: 1e-5 | $lr$: 1e-5 | $lr$: 1e-5 |
| TARGET | $lr$: 1e-5; $\lambda_{KL}$: 25; $g_{ep}$: 30 | $lr$: 1e-5; $\lambda_{KL}$: 25; $g_{ep}$: 30 | $lr$: 1e-5; $\lambda_{KL}$: 25; $g_{ep}$: 30 |
| PILoRA | $lr$: 2e-2; $lr_{pr}$: 1e-4 | $lr$: 2e-2; $lr_{pr}$: 1e-4 | $lr$: 2e-2; $lr_{pr}$: 1e-4 |
| PIP | $lr$: 5e-3 | $lr$: 5e-3 | $lr$: 5e-3 |
| LoRM | $lr$: 3e-4; $r$: 1 $\gamma$: (0, 0.5) | $lr$: 1e-4; $r$: 16 $\gamma$: (0, 0.5) | $lr$: 5e-4; $r$: 16 $\gamma$: (0, 0.5) |
| HGP | $lr$: 3e-3 | $lr$: 3e-3 | $lr$: 3e-3 |

## D.2 ImageNet-R

| $\beta$ | 0.5 | 0.1 | 0.05 |
|---|---|---|---|
| EwC | $lr$: 1e-5 | $lr$: 1e-5 | $lr$: 1e-5 |
| LwF | $lr$: 1e-5 | $lr$: 1e-5 | $lr$: 3e-5 |
| FisherAVG | $lr$: 1e-5 | $lr$: 1e-5 | $lr$: 1e-5 |
| RegMean | $lr$: 1e-5; $\gamma$: (0.1, 0.1) | $lr$: 1e-5; $\gamma$: (0.1, 0.1) | $lr$: 1e-5; $\gamma$: (0.1, 0.1) |
| CCVR | $lr$: 1e-5 | $lr$: 1e-5 | $lr$: 1e-5 |
| L2P | $lr$: 3e-2 | $lr$: 3e-2 | $lr$: 3e-2 |
| DualPrompt | $lr$: 3e-4 | $lr$: 3e-4 | $lr$: 3e-4 |
| CODA-P | $lr$: 1e-3 | $lr$: 1e-3 | $lr$: 1e-3 |
| FedProto | $lr$: 1e-5 | $lr$: 1e-5 | $lr$: 3e-5 |
| TARGET | $lr$: 1e-5; $\lambda_{KL}$: 25; $g_{ep}$: 30 | $lr$: 1e-5; $\lambda_{KL}$: 25; $g_{ep}$: 30 | $lr$: 1e-5; $\lambda_{KL}$: 25; $g_{ep}$: 30 |
| PILoRA | $lr$: 2e-2; $lr_{pr}$: 1e-4 | $lr$: 2e-2; $lr_{pr}$: 1e-4 | $lr$: 2e-2; $lr_{pr}$: 1e-4 |
| PIP | $lr$: 3e-4 | $lr$: 3e-4 | $lr$: 3e-4 |
| LoRM | $lr$: 3e-3; $r$: 2 $\gamma$: (0, 0.5) | $lr$: 1e-3; $r$: 32 $\gamma$: (0, 0.5) | $lr$: 1e-3; $r$: 16 $\gamma$: (0, 0.5) |
| HGP | $lr$: 3e-2 | $lr$: 3e-2 | $lr$: 1e-2 |

### D.3 ImageNet-A

| $\beta$ | 1.0 | 0.5 | 0.2 |
|---|---|---|---|
| EwC | *lr*: 1e-5 | *lr*: 1e-5 | *lr*: 1e-5 |
| LwF | *lr*: 1e-5 | *lr*: 1e-5 | *lr*: 3e-5 |
| FisherAVG | *lr*: 1e-5 | *lr*: 1e-5 | *lr*: 1e-5 |
| RegMean | *lr*: 1e-5; $\gamma$: (0.1, 0.1) | *lr*: 1e-5; $\gamma$: (0.1, 0.1) | *lr*: 1e-5; $\gamma$: (0.1, 0.1) |
| CCVR | *lr*: 1e-5 | *lr*: 1e-5 | *lr*: 1e-5 |
| L2P | *lr*: 3e-2 | *lr*: 3e-2 | *lr*: 3e-1 |
| DualPrompt | *lr*: 3e-2 | *lr*: 3e-2 | *lr*: 3e-2 |
| CODA-P | *lr*: 1e-2 | *lr*: 1e-2 | *lr*: 1e-2 |
| FedProto | *lr*: 3e-5 | *lr*: 1e-5 | *lr*: 1e-5 |
| TARGET | *lr*: 1e-4; $\lambda_{\text{KL}}$: 25; $g_{ep}$: 30 | *lr*: 1e-4; $\lambda_{\text{KL}}$: 25; $g_{ep}$: 30 | *lr*: 1e-4; $\lambda_{\text{KL}}$: 25; $g_{ep}$: 30 |
| PILoRA | *lr*: 2e-2; $lr_{pr}$ 1e-4 | *lr*: 1e-2; $lr_{pr}$: 1e-4 | *lr*: 2e-2; $lr_{pr}$: 1e-4 |
| PIP | *lr*: 3e-2 | *lr*: 3e-2 | *lr*: 3e-2 |
| LoRM | *lr*: 1e-2; $r$: 4 $\gamma$: (0, 0.5) | *lr*: 1e-2; $r$: 4 $\gamma$: (0, 0.5) | *lr*: 1e-2; $r$: 4 $\gamma$: (0, 0.5) |
| HGP | *lr*: 3e-2 | *lr*: 3e-2 | *lr*: 3e-2 |

### D.4 EuroSAT

| $\beta$ | 1.0 | 0.5 | 0.2 |
|---|---|---|---|
| EwC | *lr*: 1e-5 | *lr*: 1e-5 | *lr*: 1e-5 |
| LwF | *lr*: 1e-5 | *lr*: 1e-5 | *lr*: 3e-5 |
| FisherAVG | *lr*: 1e-5 | *lr*: 1e-5 | *lr*: 1e-5 |
| RegMean | *lr*: 1e-5; $\gamma$: (0.1, 0.1) | *lr*: 1e-5; $\gamma$: (0.1, 0.1) | *lr*: 1e-5; $\gamma$: (0.1, 0.1) |
| CCVR | *lr*: 1e-5 | *lr*: 1e-5 | *lr*: 1e-5 |
| L2P | *lr*: 3e-2 | *lr*: 3e-2 | *lr*: 3e-2 |
| DualPrompt | *lr*: 1e-3 | *lr*: 1e-3 | *lr*: 1e-3 |
| CODA-P | *lr*: 1e-3 | *lr*: 1e-3 | *lr*: 1e-3 |
| FedProto | *lr*: 3e-5 | *lr*: 1e-5 | *lr*: 1e-5 |
| TARGET | *lr*: 1e-5; $\lambda_{\text{KL}}$: 25; $g_{ep}$: 30 | *lr*: 1e-5; $\lambda_{\text{KL}}$: 25; $g_{ep}$: 30 | *lr*: 1e-5; $\lambda_{\text{KL}}$: 25; $g_{ep}$: 30 |
| PILoRA | *lr*: 2e-2; $lr_{pr}$: 1e-4 | *lr*: 2e-2; $lr_{pr}$: 1e-4 | *lr*: 2e-2; $lr_{pr}$: 1e-4 |
| PIP | *lr*: 1e-3 | *lr*: 1e-3 | *lr*: 1e-3 |
| LoRM | *lr*: 3e-3; $r$: 1 $\gamma$: (0, 0.5) | *lr*: 3e-3; $r$: 1 $\gamma$: (0, 0.5) | *lr*: 1e-3; $r$: 4 $\gamma$: (0, 0.5) |
| HGP | *lr*: 1e-3 | *lr*: 3e-4 | *lr*: 1e-4 |

### D.5 Cars-196

| $\beta$ | 1.0 | 0.5 | 0.2 |
|---|---|---|---|
| EwC | *lr*: 1e-5 | *lr*: 1e-5 | *lr*: 1e-5 |
| LwF | *lr*: 1e-5 | *lr*: 1e-5 | *lr*: 3e-5 |
| FisherAVG | *lr*: 1e-5 | *lr*: 1e-5 | *lr*: 1e-5 |
| RegMean | *lr*: 1e-5; $\gamma$: (0.1, 0.1) | *lr*: 1e-5; $\gamma$: (0.1, 0.1) | *lr*: 1e-5; $\gamma$: (0.1, 0.1) |
| CCVR | *lr*: 1e-5 | *lr*: 1e-5 | *lr*: 1e-5 |
| L2P | *lr*: 3e-2 | *lr*: 3e-2 | *lr*: 3e-2 |
| DualPrompt | *lr*: 1e-1 | *lr*: 1e-1 | *lr*: 1e-1 |
| CODA-P | *lr*: 3e-2 | *lr*: 3e-2 | *lr*: 3e-2 |
| FedProto | *lr*: 1e-5 | *lr*: 1e-5 | *lr*: 1e-5 |
| TARGET | *lr*: 1e-4; $\lambda_{\text{KL}}$: 25; $g_{ep}$: 30 | *lr*: 1e-4; $\lambda_{\text{KL}}$: 25; $g_{ep}$: 30 | *lr*: 1e-4; $\lambda_{\text{KL}}$: 25; $g_{ep}$: 30 |
| PILoRA | *lr*: 1e-1; $lr_{pr}$ 1e-4 | *lr*: 1e-1; $lr_{pr}$: 1e-4 | *lr*: 1e-1; $lr_{pr}$: 1e-4 |
| PIP | *lr*: 1e-1 | *lr*: 1e-1 | *lr*: 1e-1 |
| LoRM | *lr*: 1e-2; $r$: 8 $\gamma$: (0, 0.5) | *lr*: 1e-2; $r$: 8 $\gamma$: (0, 0.5) | *lr*: 1e-2; $r$: 4 $\gamma$: (0, 0.5) |
| HGP | *lr*: 3e-3 | *lr*: 3e-3 | *lr*: 3e-3 |

## D.6 CUB-200

| $\beta$ | 1.0 | 0.5 | 0.2 |
|---|---|---|---|
| EwC | $lr$: 1e-5 | $lr$: 1e-5 | $lr$: 1e-5 |
| LwF | $lr$: 1e-5 | $lr$: 1e-5 | $lr$: 3e-5 |
| FisherAVG | $lr$: 1e-5 | $lr$: 1e-5 | $lr$: 1e-5 |
| RegMean | $lr$: 1e-5; $\gamma$: (0.1, 0.1) | $lr$: 1e-5; $\gamma$: (0.1, 0.1) | $lr$: 1e-5; $\gamma$: (0.1, 0.1) |
| CCVR | $lr$: 1e-5 | $lr$: 1e-5 | $lr$: 1e-5 |
| L2P | $lr$: 3e-1 | $lr$: 3e-1 | $lr$: 3e-1 |
| DualPrompt | $lr$: 1e-1 | $lr$: 1e-1 | $lr$: 1e-1 |
| CODA-P | $lr$: 1e-3 | $lr$: 1e-3 | $lr$: 1e-3 |
| FedProto | $lr$: 1e-5 | $lr$: 1e-5 | $lr$: 1e-5 |
| TARGET | $lr$: 1e-4; $\lambda_{\mathrm{KL}}$: 25; $g_{ep}$: 30 | $lr$: 1e-4; $\lambda_{\mathrm{KL}}$: 25; $g_{ep}$: 30 | $lr$: 1e-4; $\lambda_{\mathrm{KL}}$: 25; $g_{ep}$: 30 |
| PILoRA | $lr$: 1; $lr_{pr}$ 1e-4 | $lr$: 1; $lr_{pr}$: 1e-4 | $lr$: 1; $lr_{pr}$: 1e-4 |
| PIP | $lr$: 1e-1 | $lr$: 1e-1 | $lr$: 1e-1 |
| LoRM | $lr$: 1e-2; $r$: 1 $\gamma$: (0, 0.3) | $lr$: 3e-2; $r$: 1 $\gamma$: (0, 0.3) | $lr$: 3e-2; $r$: 1 $\gamma$: (0, 0.3) |
| HGP | $lr$: 1e-1 | $lr$: 1e-1 | $lr$: 1e-1 |

# E Standard Deviations

The standard deviations across three runs for each method are presented in Table A and Table B.

Table A: Standard deviations (reported in FAA) for CIFAR-100, ImageNet-R and ImageNet-A.

| | CIFAR-100 | | | ImageNet-R | | | ImageNet-A | | |
|---|---|---|---|---|---|---|---|---|---|
| Distrib. $\beta$ | 0.5 | 0.1 | 0.05 | 0.5 | 0.1 | 0.05 | 1.0 | 0.5 | 0.2 |
| EWC | 1.38 | 2.32 | 2.67 | 1.69 | 1.14 | 1.14 | 1.34 | 0.75 | 1.38 |
| LwF | 1.15 | 2.38 | 4.25 | 1.91 | 0.93 | 1.38 | 0.30 | 0.27 | 1.61 |
| FisherAVG | 0.40 | 3.67 | 3.18 | 0.77 | 0.47 | 0.93 | 1.59 | 1.36 | 1.49 |
| RegMean | 0.01 | 1.78 | 4.87 | 1.10 | 0.85 | 1.44 | 0.59 | 1.10 | 0.70 |
| CCVR | 1.13 | 2.49 | 1.39 | 0.57 | 0.49 | 2.29 | 1.19 | 0.63 | 2.26 |
| L2P | 1.62 | 1.49 | 1.85 | 1.17 | 1.61 | 1.35 | 0.72 | 0.81 | 2.01 |
| DualPrompt | 2.17 | 3.33 | 1.42 | 0.83 | 1.68 | 1.35 | 0.34 | 1.70 | 0.86 |
| CODA-P | 1.15 | 1.96 | 0.96 | 0.98 | 1.95 | 1.19 | 0.16 | 1.63 | 0.22 |
| FedProto | 2.05 | 0.94 | 1.96 | 2.09 | 2.28 | 0.72 | 1.18 | 0.71 | 2.17 |
| TARGET | 0.79 | 1.02 | 1.81 | 0.33 | 1.58 | 1.21 | 0.65 | 0.46 | 1.34 |
| PILoRA | 0.28 | 0.98 | 4.08 | 0.29 | 1.01 | 0.33 | 0.08 | 0.31 | 0.33 |
| PIP | 0.29 | 1.32 | 0.68 | 0.52 | 0.48 | 0.34 | 0.78 | 0.20 | 0.83 |
| LoRM | 0.27 | 0.20 | 0.79 | 0.04 | 0.16 | 0.87 | 0.86 | 0.75 | 0.68 |
| HGP | 0.08 | 0.14 | 0.24 | 0.43 | 0.39 | 0.31 | 0.90 | 0.91 | 0.29 |

Table B: Standard deviations (reported in FAA) for EuroSAT, Cars-196, and CUB-200.

| Distrib. $\beta$ | EuroSAT | | | Cars-196 | | | CUB-200 | | |
|---|---|---|---|---|---|---|---|---|---|
| | 1.0 | 0.5 | 0.2 | 1.0 | 0.5 | 0.2 | 1.0 | 0.5 | 0.2 |
| EWC | 7.33 | 5.78 | 6.60 | 1.72 | 0.46 | 1.00 | 0.55 | 0.94 | 1.68 |
| LwF | 3.32 | 4.58 | 5.12 | 1.35 | 2.81 | 2.01 | 2.07 | 2.15 | 2.02 |
| FisherAVG | 5.43 | 6.07 | 3.40 | 2.10 | 1.78 | 0.78 | 0.26 | 1.35 | 2.21 |
| RegMean | 3.99 | 7.21 | 5.68 | 0.23 | 1.11 | 1.80 | 1.79 | 2.63 | 2.27 |
| CCVR | 9.01 | 7.16 | 6.26 | 1.49 | 0.87 | 2.08 | 1.22 | 1.69 | 1.73 |
| L2P | 2.52 | 3.49 | 2.02 | 0.87 | 1.94 | 0.36 | 2.38 | 0.78 | 1.21 |
| DualPrompt | 8.75 | 6.70 | 8.89 | 0.87 | 1.79 | 1.82 | 0.42 | 2.62 | 2.69 |
| CODA-P | 4.52 | 6.27 | 6.54 | 1.05 | 2.24 | 1.89 | 0.51 | 1.52 | 1.86 |
| FedProto | 3.58 | 8.74 | 7.96 | 1.11 | 0.35 | 0.87 | 0.96 | 0.67 | 1.99 |
| TARGET | 4.12 | 6.20 | 5.31 | 0.93 | 0.68 | 1.54 | 1.17 | 0.65 | 2.06 |
| PILoRA | 4.45 | 4.07 | 4.27 | 0.21 | 0.33 | 0.18 | 0.51 | 0.28 | 0.51 |
| PIP | 5.95 | 0.49 | 2.61 | 1.11 | 2.15 | 4.01 | 1.90 | 1.10 | 1.42 |
| LoRM | 1.75 | 3.34 | 6.51 | 1.07 | 0.26 | 0.92 | 0.86 | 0.98 | 0.44 |
| HGP | 1.03 | 1.20 | 1.37 | 1.73 | 1.58 | 0.30 | 0.36 | 0.19 | 0.60 |

## F Additional Figures: Federated Bias

In this section, we show the relationship between response entropy and performance on two additional datasets (Figure A).

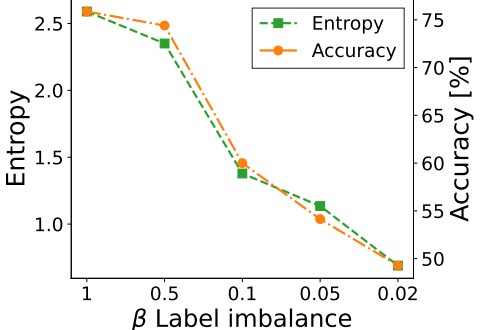 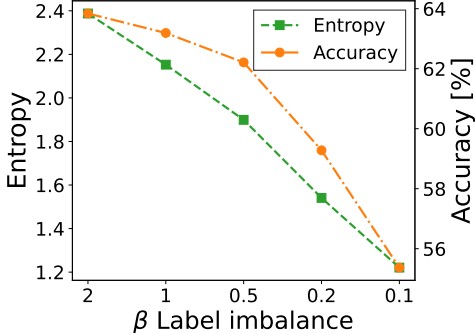

Figure A: **Federated bias.** Entropy of the response histograms, averaged on all clients, compared with FL performance, for ImageNet-R (left) and ImageNet-A (right). The experimental setting follows Section 2.

## G Ablation study: Forgetting

To evaluate the efficacy of our proposed methodology, we perform a granular assessment of model performance across tasks. Specifically, we analyze the accuracy trends of previously learned tasks in comparison to the performance of the current task. This dual-metric approach allows us to closely monitor the forgetting phenomenon. By visualizing these trajectories, we show HGP's prompting methodology preserves prior knowledge while successfully integrating new classes (see Figures B to D).

## H Ablation Study: Privacy and Defense Resilience

In this section, we evaluate the privacy-preserving properties of our framework by subjecting it to membership inference attacks across three datasets: ImageNet-A, EuroSAT, and CIFAR-100. Our results demonstrate

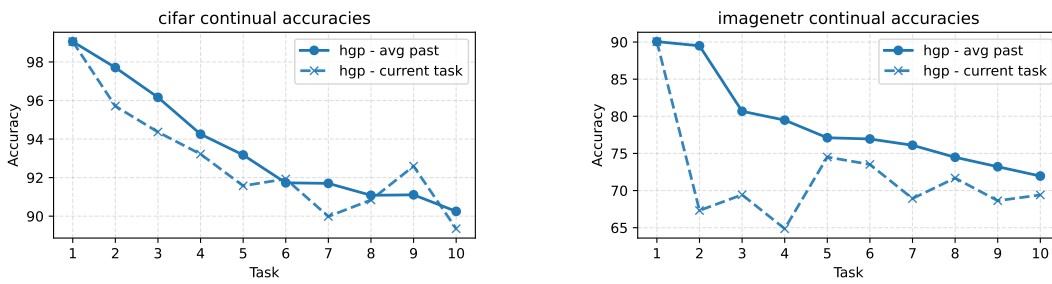

Figure B: Accuracies on past vs. current tasks for CIFAR-100 (left) and ImageNet-R (right).

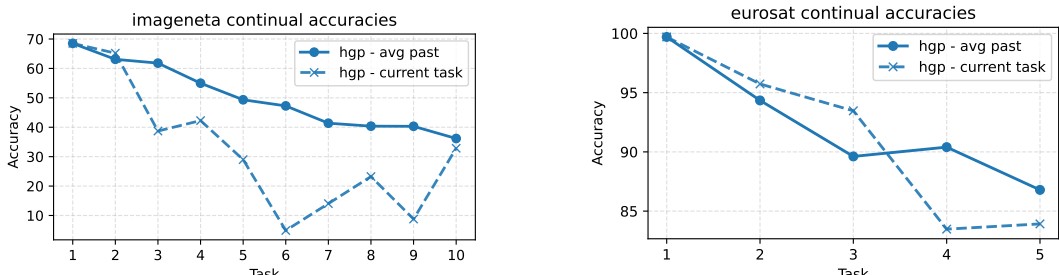

Figure C: Accuracies on past vs. current tasks for ImageNet-A (left) and EuroSAT (right).

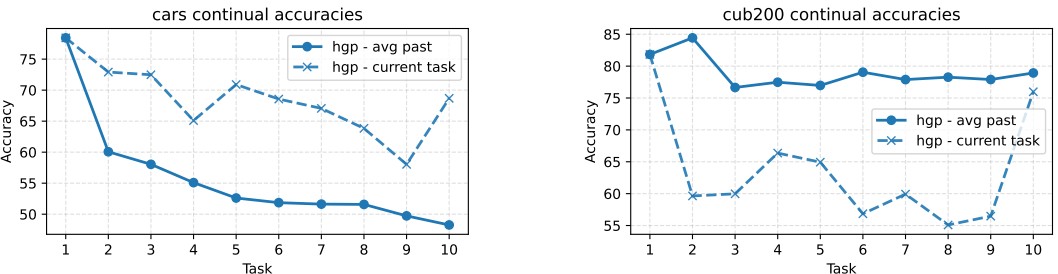

Figure D: Accuracies on past vs. current tasks for Cars-196 (left) and CUB200 (right).

that the framework exhibits strong resistance to membership leakage, especially under strict false-positive constraints (see Figures E to G).

## H.1 Attacker Capability and Evaluation Metrics

Across all experiments, we assume a strong threat model where the attacker is provided with a released client/class prototype (mean feature vector and diagonal covariance) and candidate same-class feature vectors. The attacker scores each candidate against the prototype using Gaussian log-likelihood, Mahalanobis distance, or Cosine similarity to infer target client origin. It is worth noting that, since negatives are same-class samples from other clients, the attacks strictly measure client-membership leakage rather than general class recognition.

To demonstrate HGP's privacy guarantees, performance is evaluated using metrics where lower scores indicate successful defense:

- **AUROC / Weighted AUROC:** measures ranking quality. Values approaching 0.5 indicate that HGP has successfully forced the attacker into random guessing. Positive-weighted AUROC reduces the influence of smaller prototypes.

- **AUPRC / AUPRC Lift:** evaluates precision-recall. Since baseline AUPRC naturally matches the class prevalence (positive rate), we look at AUPRC lift (AUPRC/positive rate). A lift near 1.0 means that the attacker fails to gain any meaningful advantage over the baseline distribution.

- **TPR@1%FPR:** measures true member recovery under a 1% false-positive budget. This represents the most realistic operational constraint for an attacker: low values confirm that HGP is safe.

## H.2 ImageNet-A

This filtered set includes 47 client/class prototypes, establishing a baseline positive rate (prevalence) of 0.235. This represents our most challenging scenario due to the relatively low support threshold (average of ∼15 positives). As shown in Table C, while the attacker gains some ranking traction in this limited-data regime (Cosine AUROC 0.846), the defense remains practically effective under realistic constraints. Importantly, under a strict 1% false-positive budget (TPR@1%FPR), the attacker still completely fails to identify 70% to 78% of true members on average. This indicates that even in worst-case, low-support scenarios, definitive membership extraction remains unreliable.

Table C: ImageNet-A Defense Summary. Despite some ranking signal, strict extraction (TPR@1%FPR) remains difficult.

| Attack | Mean AUROC | Weighted AUROC | Mean AUPRC | Mean Lift | TPR@1%FPR |
|---|---|---|---|---|---|
| Gaussian | 0.762 | 0.763 | 0.542 | 2.522 | 0.220 |
| Mahalanobis | 0.762 | 0.763 | 0.542 | 2.522 | 0.220 |
| Cosine | 0.846 | 0.843 | 0.647 | 3.036 | 0.301 |

## H.3 EuroSAT

This set evaluates well-supported prototypes, with a mean of ∼160 positives and a baseline positive rate of 0.116. Results in Table D demonstrate robust defense against membership inference. HGP effectively neutralizes the attack, degrading the mean AUROC to 0.541–0.558, which is nearly indistinguishable from random guessing. Furthermore, the mean AUPRC (0.154) offers only a marginal lift over the baseline, and the strict TPR@1%FPR metric is very low (∼0.034). This proves that an attacker cannot reliably identify members in well-supported classes without being overwhelmed by false positives.

Table D: EuroSAT Defense Summary. HGP reduces attacker metrics to near-random levels.

| Attack | Mean AUROC | Weighted AUROC | Mean AUPRC | Mean Lift | TPR@1%FPR |
|---|---|---|---|---|---|
| Gaussian | 0.558 | 0.546 | 0.154 | 1.439 | 0.034 |
| Mahalanobis | 0.558 | 0.546 | 0.154 | 1.439 | 0.034 |
| Cosine | 0.541 | 0.535 | 0.144 | 1.346 | 0.027 |

## H.4 CIFAR-100

This configuration confirms the privacy-preserving nature of the model for adequately supported prototypes (mean positive rate of 0.166).

Table E highlights the consistent failure of the attacks to extract meaningful membership data. The absolute ranking signal is heavily suppressed (Mean AUROC 0.551–0.569), and the AUPRC Lift remains low. Most notably, at a 1% FPR budget, the attacker's success rate is negligible, recovering only 2.4% to 2.5% of true members. These results confirm that for adequately supported prototypes, membership privacy is strongly protected against sophisticated statistical attacks.

Table E: CIFAR-100 Defense Summary. Attacks fail to recover meaningful membership signal under strict budgets.

| Attack | Mean AUROC | Weighted AUROC | Mean AUPRC | Mean Lift | TPR@1%FPR |
|---|---|---|---|---|---|
| Gaussian | 0.569 | 0.541 | 0.198 | 1.372 | 0.025 |
| Mahalanobis | 0.569 | 0.541 | 0.198 | 1.372 | 0.025 |
| Cosine | 0.551 | 0.536 | 0.197 | 1.351 | 0.024 |

## I  On the Diagonal Covariance Assumption

To ensure efficiency and numerical stability, we parameterize each generative prototype $\mathcal{N}_{m,c}$ with a *diagonal* covariance matrix (stored as a vector). This choice is well motivated for the feature space we operate in. First, the Neural Collapse phenomenon (Papyan et al., 2020) shows that intra-class variability collapses towards an isotropic state in the terminal phase of training, so off-diagonal correlations carry little additional class-conditional information; consistently, recent work shows that the InfoNCE objective used by many pre-trained backbones induces an isotropic Gaussian feature structure (Betser et al., 2026). Second, a full covariance in the embedding space of a transformer-based pre-trained model requires estimating a huge number of parameter per class (*e.g.*, feature_dim $= 768$ for ViT-B/16 $\rightarrow d(d+1)/2 \approx 294{,}000$ parameters), which makes the problem ill-posed on datasets with as few as $\sim$20 samples per class and therefore requires heavy regularization to remain invertible. Empirically, Table F shows that a (Tikhonov-regularized) full covariance yields no consistent accuracy gain over the diagonal parameterization, while incurring higher computation an memory costs. We further verify in Section J that the diagonal assumption does not cause semantic collapse of the synthetic features.

Table F: **Diagonal vs. full covariance.** FAA (reported for $\beta$ equal to the least imbalanced setting of each dataset) of HGP when each generative prototype $\mathcal{N}_{m,c}$ uses a diagonal covariance (default) or a full covariance matrix estimated with strong Tikhonov regularization. The two variants are comparable across all datasets, while the full covariance incurs substantially higher memory and compute.

| Covariance | CIFAR-100 | ImageNet-R | ImageNet-A | EuroSAT | Cars-196 | CUB-200 |
|---|---|---|---|---|---|---|
| Diagonal (ours) | 90.39 | 72.64 | 41.61 | 87.97 | 52.13 | 79.71 |
| Full (Tikhonov) | 90.03 | 72.77 | 40.11 | 88.53 | 51.73 | 78.89 |

## J  Visualization of Feature Distributions

To validate our choice of feature distribution modeling using diagonal covariance matrices, we provide qualitative visualizations of the 768-dimensional ViT feature embeddings. Specifically, we compare the distributions of real ViT features against the synthetic features generated by our diagonal Gaussian prototypes. We utilize both t-SNE and UMAP for dimensionality reduction. To ensure the robustness of our analysis across diverse visual domains, we perform these projections on three datasets: CIFAR-100, ImageNet-A, and EuroSAT.

As illustrated in Figures E to G, the synthetic features generated via the diagonal assumption closely align with the manifold of the real features across all three datasets. We note that the semantic structure is preserved; intra-class cohesion and inter-class separation are well-maintained in the synthetic distributions.

These visualizations empirically support our conclusion that the diagonal approximation effectively captures the necessary class-conditional variance in this high-dimensional regime. It serves as a robust regularizer capable of generating representative prototypes without explicitly modeling full feature covariances.

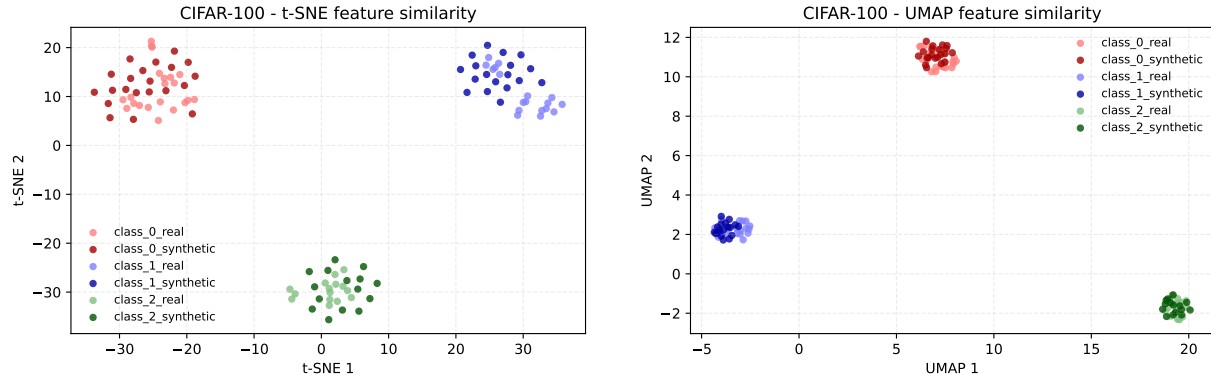

Figure E: **Feature visualizations for CIFAR-100.** t-SNE (left) and UMAP (right) projections comparing real ViT features with synthetic features generated via diagonal covariance matrices.

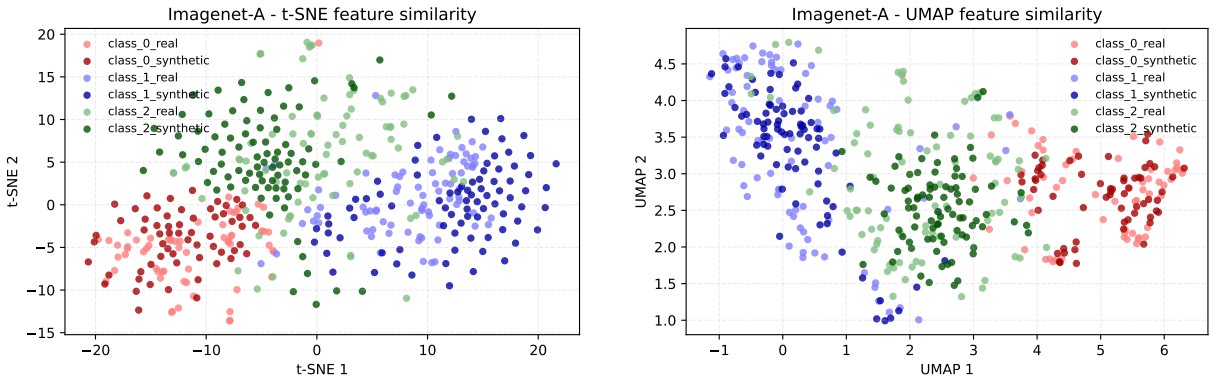

Figure F: **Feature visualizations for ImageNet-A.** t-SNE (left) and UMAP (right) projections comparing real ViT features with synthetic features generated via diagonal covariance matrices.

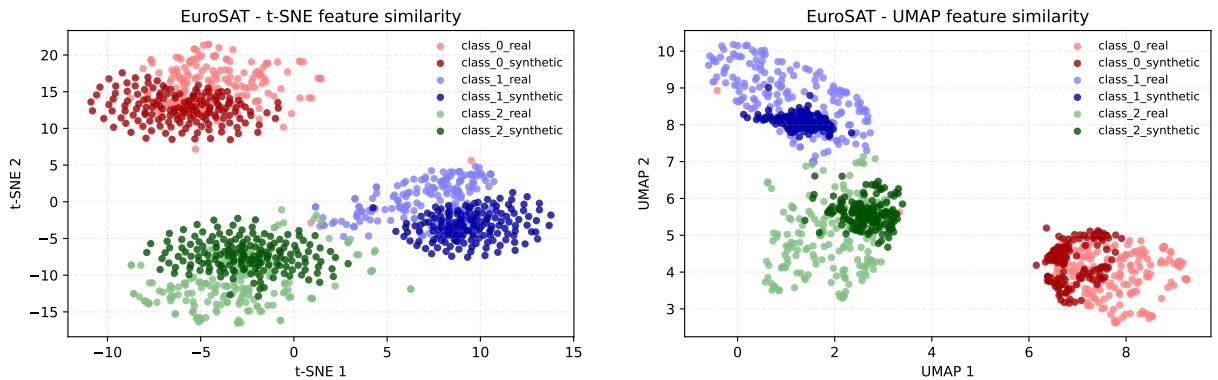

Figure G: **Feature visualizations for EuroSAT.** t-SNE (left) and UMAP (right) projections comparing real ViT features with synthetic features generated via diagonal covariance matrices.

