# OpenReview forum: "Federated Class-Incremental Learning with Hierarchical Generative Prototypes"
_TMLR — Decision pending for TMLR_

### Review · Reviewer_ULz2 · 2026-03-31

**Summary Of Contributions:**

The paper studies Federated Class-Incremental Learning (FCIL) and emphasizes that Incremental Bias and Federated Bias are the key causes of performance degradation in FCIL. The authors then propose a method called Hierarchical Generative Prototypes (HGP), which uses prefix-tuning with a pre-trained ViT to confine both biases to the final classification layer and rebalances the global classifier on the server side by sampling synthetic features from a hierarchical GMM constructed from prototypes across clients. Experiments on 6 datasets demonstrate the state-of-the-art performance of HGP.

**Audience:**

Yes

**Audience Explanation:**

Federated Class-Incremental Learning (FCIL) is an active research area with growing interest. FCIL addresses a practical and challenging scenario where models must handle both decentralized, non-IID data and evolving class distributions over time—factors that are highly relevant to real-world deployments. This paper proposes a simple yet effective method that achieves state-of-the-art performance. Researchers working on federated learning, continual learning, or prompt-tuning-based adaptation would find the results and analysis useful.

**Broader Impact Concerns:**

A Broader Impact Statement is missing from the current paper. Because the method requires sharing feature statistics with a central server, the authors should probably include a quick discussion on privacy. It would be helpful to know if there are other vulnerabilities to consider beyond the feature inversion cases covered in Section 4.6.

**Claims And Evidence:**

Yes

**Claims Explanation:**

- The experimental results are comprehensive, encompassing 6 datasets, 3 levels of heterogeneity, and 13 baselines across class-incremental learning, federated learning, model merging, and FCIL-specific methods. The ablation study in Table 3 effectively validates the impact of each component of the proposed HGP framework. Furthermore, the feature bias analysis in Figure 3 provides reasonable support for the prompting-based design choice.

- Regarding the claim in the contribution summary that federated bias is the primary cause of performance degradation in FCIL, there is insufficient empirical evidence. The entropy-accuracy analysis in Section 2 is limited to CIFAR-100 with only 3 $\alpha$ values. Increased heterogeneity exacerbates multiple factors simultaneously, including federated bias, catastrophic forgetting, and aggregation errors. To justify the "primary cause" claim, I suggest the authors either provide a controlled experiment isolating federated bias from other confounding factors or demonstrate the correlation consistently across a wider range of datasets.

**Requested Changes:**

#### Critical changes
- Justification of the "primary cause" claim: The paper identifies bias as "the primary cause" of performance degradation in FCIL. However, the evidence in Section 2 is currently limited to an entropy-accuracy correlation across only three $\alpha$ values on CIFAR-100. To support such a strong claim, I would recommend either extending this analysis to a broader range of datasets or tempering the language (e.g., "a major factor").

- Inconsistency in experimental settings: There is a discrepancy regarding the $\alpha$ values used for the Cars-196 dataset. Section 4.1 and Appendix C report $\alpha \in \{1.0, 0.5, 0.2\}$, whereas Table 2 lists $\alpha \in \{0.5, 0.1, 0.05\}$. Could the authors please clarify which configuration was used for the final results?

#### Suggestions for improvement
- Per-task accuracy: The paper would benefit from reporting per-task accuracy curves. Given the focus on Incremental Bias, visualizing how the performance on "old" versus "new" classes evolves across incremental steps would provide much more direct support for the proposed method's effectiveness.

- Rigorous privacy analysis: The privacy-preserving claims in Section 4.6 currently feel a bit limited. The authors provide only one qualitative example on Tiny-ImageNet (which isn't part of the main experiments) and lack quantitative metrics. A more thorough analysis or a discussion of other potential attack techniques (e.g., membership inference or more advanced inversion attacks) would significantly strengthen this section.

- Hyperparameter tuning anaysis: The hyperparameter search strategy is not explicitly described. Given the varying configurations across different methods and datasets in Appendix B, it would be helpful to clarify if the baselines received a comparable level of tuning effort to ensure a fair and balanced comparison.

---

> ### Author Response · Authors · 2026-05-04
>
> We sincerely thank the reviewer for their constructive feedback and for recognizing the quality of our experimental analysis and the effectiveness of our method.
>
> ### On the "primary cause" claim
>
> In response to the reviewer's feedback, we have refined the Abstract and Introduction to characterize Federated Bias and Incremental Bias as "major factors" rather than the primary cause of performance degradation. Furthermore, we have expanded our analysis in the Appendix to include a wider range of datasets and varying Dirichlet label imbalance values ($\beta$). These additional studies consistently support our initial thesis across different experimental settings.
>
>
> ### Inconsistency in Experimental Settings (Cars-196)
>
> We apologize for the typografical error in the documentation. The experiments for Cars-196 were conducted using $\beta \in \{1.0, 0.5, 0.2\}$, as reported in Section 4.1 and Appendix C. We have rectified these values in the revised manuscript to ensure consistency.
>
> ### Per-task accuracies
>
> We thank the reviewer for their suggestion to visualize the evolution of old vs. new class performance to evaluate Incremental Bias.
>
> In our revision, we have added Figures B, C, and D to the Appendix, showing the per-task accuracy curves for all datasets. The results demonstrate that HGP effectively mitigates catastrophic forgetting; notably, performance on previous tasks remains robust, frequently surpassing the accuracy of the current task:
>
> |Dataset|Avg acc. (past)|Avg acc. (current)|
> |-|-|-|
> |CIFAR-100|**92.69**|91.84|
> |ImageNet-R|**77.73**|69.82|
> |ImageNet-A|**47.63**|28.10|
> |EuroSAT|**89.79**|88.65|
> |Cars-196|50.17|**64.47**|
> |CUB200|**78.13**|61.21|
>
>
> ### Additional privacy analysis
>
> We agree with the reviewer that the inversion attack we presented is only qualitative. Therefore, we have added a quantitative privacy assessment in Section H (Appendix) using Membership Inference Attack (MIA) metrics. The results demonstrate that sharing our prototypes (i.e., aggregated Gaussian statistics) results in low membership leakage.
>
> ### On hyperparameters search
>
> To ensure a fair comparison, we dynamically adjust each hyperparameter search range for each methodology so that the optimal configuration was always bracketed by other tested values. We have better specified our strategy in the Hyperparameters section in the Appendix.
>
> ### On entropy-accuracy analysis
>
> We have updated Figure 1 testing on more $\beta$ label imbalance settings for CIFAR-100, and included additional experiments on other datasets in Section F.

---

### Review · Reviewer_VPJr · 2026-04-04

**Summary Of Contributions:**

This work shed light on the importance of Incremental Bias and Federated Bias in Federated Continual Learning (FCL). The authors propose an approach that constrains both biases to the last layer by efficiently fine-tuning a pre-trained backbone using learnable prompts, resulting in clients that produce less biased representations and more biased classifiers. The proposed methodology are validated across six datasets, each including three different scenarios.

**Audience:**

Yes

**Audience Explanation:**

Researchers from the community of federated learning are likely to be interested in knowing the findings of this paper. However, the significance of the findings is questionable. The technical contributions proposed seem incremental.

**Broader Impact Concerns:**

N.A. There are no concerns on the ethical implications of the work that would require adding a Broader Impact Statement.

**Claims And Evidence:**

Yes

**Claims Explanation:**

The claims made in the submission are mostly supported by accurate, convincing, and clear evidence. However, the benchmarks used in the experiments are small-scale benchmarks. More experiments on larger-scale benchmarks will further strengthen the experimental validation of this work.

**Requested Changes:**

1. The figures, especially Fig.2, should be polished for better presentation.

2. More discussions and experimental validations on generalization to training-from-scratch scenarios should be added.

---

> ### Author Response · Authors · 2026-05-04
>
> We sincerely thank the reviewer for their time and constructive feedback. We are glad they found our claims to be supported by accurate and clear evidence across our experimental scenarios. Below, we address the raised concerns.
>
> ### On the technical contribution
> While we understand the concern that applying a new prompt-tuning method to a pre-trained backbone might initially appear incremental, we respectfully emphasize that the core novelty of our work lies in the simultaneous analysis and mitigation of both Incremental and Federated biases. Our approach does not merely apply a **novel prompting methodology**; rather, **it leverages prompting strategically to constrain** these dual biases to the final layer. This deliberate design choice **makes the subsequent rebalancing procedure vastly more effective**. As a result, this framework achieves state-of-the-art performance, outperforming existing baselines (including rebalancing methods like CCVR) by a substantial margin.
>
> ### On the generalization to training from scratch scenarios
> We appreciate the reviewer's suggestion to explore training-from-scratch scenarios. However, our proposed methodology is built on prompt-tuning: a fine-tuning approach that inherently necessitates a pre-trained backbone. Without prompt-tuning, our core mechanism constraining both Incremental and Federated biases to the last layer collapses.
>
> We explicitly acknowledged this architectural constraint in the Limitations section of our submission, emphasizing that our findings may not generalize to settings where models are trained from scratch. Such a scenario falls outside the theoretical scope of our work.
>
> ### Polishing Figure 2
> We have polished Figure 2 in our revision, to improve its clarity and overall presentation. The new version should have a more intuitive visual flow.

---

### Review · Reviewer_6Gbg · 2026-04-20

**Summary Of Contributions:**

The authors propose Hierarchical Generative Prototypes (HGP) for Federated Class-Incremental Learning (FCIL). The method attempts to address two phenomena the authors term "Incremental Bias" (IB) and "Federated Bias" (FB). To mitigate these, the authors employ a single shared learnable prompt within the early blocks of a pre-trained Vision Transformer (ViT) to constrain feature drift, alongside a server-side classifier rebalancing technique that samples from hierarchical Gaussian Mixture Models (GMMs) representing client-class feature distributions. The authors evaluate their approach on six datasets (CIFAR-100, ImageNet-R, ImageNet-A, EuroSAT, Cars-196, CUB-200) and compare it against various Continual Learning, Federated Learning, and FCIL baselines.

While the problem of FCIL is highly relevant, the execution of the proposed solution contains significant flaws. The paper conflates standard label skew with novel bias definitions, relies on a highly constrained prompt mechanism that limits model plasticity, and presents an ablation study that inadvertently undermines the core claims of the methodology.

**Audience:**

Yes

**Audience Explanation:**

The intersection of Federated Learning and Class-Incremental Learning is a highly challenging and practically important domain. TMLR readers interested in distributed continuous learning, privacy-preserving rehearsal mechanisms, and PEFT would find the premise of the paper relevant, even if the current methodology and empirical justifications require substantial revision.

**Broader Impact Concerns:**

The authors provide a brief privacy analysis in Section 4.6 regarding feature inversion. However, while feature inversion might fail to recover exact visual semantics (Figure 5), generative prototypes inherently capture the statistical distribution of local data. In highly sensitive domains (e.g., medical imaging), transmitting class-conditional distributions could still leak aggregate patient demographics or site-specific anomalies. A brief acknowledgement of distribution-level privacy leakage should be added to the limitations.

**Claims And Evidence:**

No

**Claims Explanation:**

The core claims of the paper are not supported by the evidence provided, due to significant theoretical and empirical weaknesses in the following areas:

1. The paper claims to "shed light" on "Federated Bias" (FB). However, in Section 2 (Figure 1), the authors measure FB by evaluating the entropy of local model predictions under extreme Dirichlet label skew ($\beta=0.05$). The observation that models trained on highly skewed non-IID data predict their majority classes more frequently is a well-established foundational fact of Federated Learning, generally categorized under client drift or simple statistical heterogeneity. Rebranding this basic phenomenon as a novel "Federated Bias" does not provide new theoretical insights. Furthermore, the claim in Section 4.3 (Figure 3) that prompt tuning "produces smaller feature bias" than full fine-tuning is inherently obvious. When freezing a massive pre-trained ViT and only tuning ~1.5% of the parameters via prompts, the feature space will naturally drift less than when updating 100% of the network. This is a default characteristic of Parameter-Efficient Fine-Tuning (PEFT), not a unique triumph of the proposed method.

2.  Prompt Plasticity: In Section 3.2 ("Prompting"), the authors state they learn a single shared prompt across all tasks, contrasting this with standard CIL methods that use prompt pools (e.g., L2P, CODA-Prompt). While this saves a forward pass during inference, a single shared prompt fundamentally lacks the capacity to absorb heterogeneous, sequential task knowledge without severe catastrophic forgetting. The prompt is inevitably overwritten by subsequent tasks.Feature Distribution Modeling: In Section 3.2 and later clarified in Section 4.5 ("Computational Complexity"), the authors use diagonal covariance matrices to parameterize their generative prototypes ($\mathcal{N}_{m,c}$). Modeling high-dimensional (768-d) ViT feature embeddings as independent, uncorrelated variables via diagonal Gaussians strips away the rich semantic correlations that self-attention mechanisms build. The authors claim in Section 4.5 that "full covariance matrices provide no accuracy gains," but this likely points to an underlying issue with how the feature space is being structured or utilized, rather than validating the diagonal assumption.

3.  The ablation study in Section 4.3 (Table 3) inadvertently dismantles the paper's core narrative. The authors argue that prompting "confines both types of bias." However, Table 3 shows that using only the prompt without Classifier Rebalancing (CR) yields an abysmal 52.29% FAA on CIFAR-100. Adding $CR_{old}$ jumps the accuracy to 81.91%. Crucially, the baseline without prompting but with $CR_{our}$ achieves 85.43% on CIFAR-100. This indicates that the heavy lifting of the performance is entirely driven by the server-side generative replay (GMM sampling), not the prompt design. The prompting mechanism appears to be a marginal add-on rather than a synergistic core component.

4. While the authors cite general CIL prompting methods (Wang et al., 2022a; 2022b), the related work (Section 5) fails to engage with the rapidly expanding literature on continuous prompt tuning that specifically addresses domain shifts and dynamic feature alignment. Furthermore, the discussion of prototype-based methods ignores recent advancements in contrastive prototype alignment, making the proposed simple GMM approach feel somewhat dated.

**Requested Changes:**

1. Remove the claims of discovering or formalizing "Federated Bias" as a novel concept in Section 1 and Section 2. Frame the work instead around mitigating standard non-IID label skew and forgetting via PEFT and generative replay.

2. Provide empirical evidence showing how a single shared prompt (Section 3.2) evolves over $T$ tasks. Show the cosine similarity of the prompt parameters between Task 1 and Task 10. If the prompt suffers from extreme forgetting, acknowledge that the backbone is doing all the work.

3. The authors must provide a clearer explanation for why prompting alone performs so poorly, and include an additional row in Table 3: Full Fine-tuning + Both CR mechanisms. This is necessary to prove whether the prompt actually provides a ceiling performance increase, or if standard fine-tuning with the exact same replay buffer achieves identical results.

4. Provide a t-SNE or UMAP visualization of the synthetic features generated by the diagonal Gaussians (Section 3.2) compared to the real features. Demonstrate that the diagonal covariance assumption does not result in semantic collapse in the 768-d space.

5. Update Section 5 to include and contrast against recent continuous prompt learning strategies and advanced federated prototype aggregation methods.

---

> ### Author Response · Authors · 2026-05-04
>
> We thank reviewer 6Gbg for their feedback.
> ## Federated Bias claims
> *Client drift* is a term introduced in Karimireddy, 2021, for a phenomenon known as *Weight Divergence*, previously studied in Zhao, 2018. Both terms quantify the **difference in parameter space** from two different training with the same initialization. What we call Federated Bias was introduced in Luo, 2021, as "bias". **It is defined as difference in the clients representations**. We cite the original source in our introduction and compare against their method (CCVR) in our experiments. **Our contribution is not to re-brand weight divergence, but to investigate Federated Bias and propose an effective methodology to tackle it**.
> ## Prompting and feature bias reduction
> We agree that, as written in our manuscript, ideally prompting reduces feature bias. In the absence of other works that state the same, we follow standard research practice and validate this experimentally in Section 4.3.
> ## Prompt plasticity
> Since our approach employs a wider prompt than those used by CODA or L2P, its capacity allows learning with limited forgetting without the need for routing mechanisms. Empirically, HGP usually maintains higher accuracy on past tasks:
> |Dataset|Avg acc. (past)|Avg acc. (current)|
> |-|-|-|
> |C-100|**92.69**|91.84|
> |IN-R|**77.73**|69.82|
> |IN-A|**47.63**|28.10|
> |eSAT|**89.79**|88.65|
> |CARS|50.17|**64.47**|
> |CUB|**78.13**|61.21|
>
> We added the related plots in the Appendix (Figures B, C, D).
> ## Feature distribution modeling
> We respectfully disagree that a diagonal covariance assumption indicates a methodological flaw. First, the Neural Collapse literature (Papyan, 2020) demonstrates that intra-class variability collapses into an isotropic state at convergence. Second, estimating a full covariance matrix in a $768$-dimensional space (ViT-B/32) requires fitting roughly $294000$ parameters per class, which becomes an ill-posed problem on downstream datasets with limited per-class samples (as low as $20$), hence requiring a **very strong Tikhonov reg. or a diagonal covariance matrix**. Also, recent work (Betser, 2026) shows that the InfoNCE loss (used in many pre-trained ViTs) induces an isotropic Gaussian feature structure. Also, we report HGP's performance with diagonal and full covariance (with strong Tikhonov reg.), showing they are comparable:
> |Dataset: $\beta = 0.5$|Diagonal cov.|Full cov.|
> |-|-|-|
> |CIFAR-100|90.39|90.03|
> |ImageNet-R|72.64|72.77|
> |ImageNet-A|41.61|40.11|
> |EuroSAT|87.97|88.53|
> |Cars-196|52.13| 51.73|
> |CUB200|79.71|78.89|
> ## On HGP components ablations
> We believe this critique stems from a misunderstanding of our motivation and Table 3:
>  - First, we state that prompting "confines both types of bias **to the final classification layer**". This is what makes Classifier Rebalancing effective; they are **designed to work together, not as competing components**.
>  - Second, we believe the reviewer misidentifies the components in Table 3, as **the $CR_{old}$ row already incorporates prompting**.
>  - Finally, if performance was solely due to generative replay, our method would perform as good as the synthetic replay baseline (CCVR). Instead, HGP outperforms CCVR across all settings, **achieving accuracy gains of $>15\%$ on half of the datasets**. Such a large gap underlines that our prompting is not a "marginal add-on" but a critical component that makes replay effective.
>
> Requested changes:
> 1. We did not claim to formalize Federated Bias but we cite the original source (Luo, 2021) in our introduction, and compare against CCVR in our experimental section. We made this clearer in our revision.
> 2. We hope the reviewer appreciates our additional experiment on HGP's forgetting within this rebuttal and Section E (revision).
> 3. Prompting performs poorly because it is designed to **increase Federated Bias** within the network. It works **together with the rebalancing strategy**, not alone by itself. We included full fine-tuning + rebalancing results in our revision.
> 4 & 5. We made the requested changes in our revision.
> ## Broader impact
> While our feature inversion analysis (Figure 5) demonstrates resilience to data recovery, transmitting feature statistics could leak membership of original features. We added this to our limitations, and included an analysis on Membership Inference attacks in Section H.
> - Karimireddy, S. P., et al. (2021). Scaffold: Stochastic controlled averaging for federated learning. In ICML.
> - Zhao, Y., et al. (2018). Federated learning with non-iid data.
> - Luo, M., et al. (2021). No fear of heterogeneity: Classifier calibration for federated learning with non-iid data. In NeurIPS.
> - Papyan, V., et al (2020). Prevalence of neural collapse during the terminal phase of deep learning training. In NAS.
> - Betser, R., et al. (2026). InfoNCE Induces Gaussian Distribution. In ICLR. Oral presentation.
> - Mi Luo, et al. (2021). No fear of heterogeneity: Classifier calibration for federated learning with non-iid data. In NeurIPS.

---

> > ### Comment · Reviewer_6Gbg · 2026-05-05
> >
> > Thank you for the detailed response. I am satisfied that the main concerns raised in my review have been addressed. I encourage the authors to incorporate the key clarifications and additional analyses from the rebuttal into the revised manuscript.

---

> > > ### Author Response · Authors · 2026-05-06
> > > **Official comment by Authors**
> > >
> > > Dear Reviewer 6Gbg,
> > >
> > > Thank you for your positive assessment and for the constructive feedback throughout the discussion.
> > > We are glad that the rebuttal addressed your main concerns. In the revised manuscript, we will incorporate the key clarifications and additional analyses discussed in our response.

---

### Decision · Action_Editor_exYy · 2026-06-10

**Recommendation:** Accept with minor revision

**Additional Comments:**

In the final version of the manuscript, I encourage the authors to incorporate the key clarifications and additional analyses discussed during the review process, particularly those highlighted by Reviewers ULz2 and 6Gbg. Doing so will further strengthen the paper and improve its accessibility to readers.

**Audience:**

Yes

**Audience Explanation:**

Even if some reviewers expressed concern on the novelty of the contribution, novelty is not the main concern in TMLR. In my opinion, the work falls well within TMLR's acceptance criteria which is presenting sound and clearly communicated research that is likely to be of interest to the TMLR audience.

**Claims And Evidence:**

Yes

**Claims Explanation:**

Upon submission, the reviewers raised concerns regarding the soundness of certain aspects of the paper, particularly the presentation and framing of its main contributions. Nevertheless, the authors have made a substantial effort to address the reviewers' comments and questions. Overall,  after revision, it seem to me like the paper would provide an interesting contribution to the TMLR community, and its claims are now supported by reasonably convincing empirical evidence.

---

> ### Author Response · Authors · 2026-06-14
> **Response to Decision**
>
> Dear Action Editor,
>
> Thank you very much for your positive decision and for handling our submission throughout the review process.
>
> We are grateful to you and to the reviewers for the constructive feedback and for the time dedicated to evaluating our work. We will carefully revise the manuscript to incorporate the key clarifications and additional analyses discussed during the review process, in particular those highlighted by Reviewers ULz2 and 6Gbg, as suggested in your decision.
>
> Thank you again for your guidance.
>
> Best regards,
> The Authors